# The C9orf72 protein interacts with Rab1a and the ULK1 complex to regulate initiation of autophagy

Christopher P Webster[1,‡], Emma F Smith[1,‡], Claudia S Bauer[1], Annekathrin Moller[1], Guillaume M Hautbergue[1], Laura Ferraiuolo[1], Monika A Myszczynska[1], Adrian Higginbottom[1], Matthew J Walsh[1], Alexander J Whitworth[2,†], Brian K Kaspar[3], Kathrin Meyer[3], Pamela J Shaw[1], Andrew J Grierson[1,§] & Kurt J De Vos[1,§,*]

## Abstract

A GGGGCC hexanucleotide repeat expansion in the *C9orf72* gene is the most common genetic cause of amyotrophic lateral sclerosis and frontotemporal dementia (C9ALS/FTD). *C9orf72* encodes two C9orf72 protein isoforms of unclear function. Reduced levels of C9orf72 expression have been reported in C9ALS/FTD patients, and although C9orf72 haploinsufficiency has been proposed to contribute to C9ALS/FTD, its significance is not yet clear. Here, we report that C9orf72 interacts with Rab1a and the Unc-51-like kinase 1 (ULK1) autophagy initiation complex. As a Rab1a effector, C9orf72 controls initiation of autophagy by regulating the Rab1a-dependent trafficking of the ULK1 autophagy initiation complex to the phagophore. Accordingly, reduction of C9orf72 expression in cell lines and primary neurons attenuated autophagy and caused accumulation of p62-positive puncta reminiscent of the p62 pathology observed in C9ALS/FTD patients. Finally, basal levels of autophagy were markedly reduced in C9ALS/FTD patient-derived iNeurons. Thus, our data identify C9orf72 as a novel Rab1a effector in the regulation of autophagy and indicate that C9orf72 haploinsufficiency and associated reductions in autophagy might be the underlying cause of C9ALS/FTD-associated p62 pathology.

**Keywords** amyotrophic lateral sclerosis; autophagy; C9orf72; frontotemporal dementia; Rab GTPase

**Subject Categories** Neuroscience

**The EMBO Journal (2016) 35: 1656–1676**

## Introduction

Macroautophagy (hereafter termed autophagy) is a conserved lysosomal degradation pathway that is an essential process to maintain cellular homeostasis. A double lipid membrane, the phagophore, engulfs cellular components targeted for degradation including proteins and organelles. The resulting autophagosome fuses with lysosomes, and its content is degraded to release cell components that can be reused. Autophagy is induced in response to starvation to boost nutrient availability, but is also responsible for the removal of defunct, damaged organelles and misfolded/aggregated proteins. Defects in autophagy are linked with several human diseases, including neurological disorders such as Parkinson's disease and amyotrophic lateral sclerosis (ALS) (Harris & Rubinsztein, 2012).

A GGGGCC hexanucleotide repeat expansion in intron 1 of the *C9orf72* gene was found to be a common cause of both ALS and frontotemporal dementia (FTD) (DeJesus-Hernandez *et al*, 2011; Renton *et al*, 2011), accounting for approximately 40 and 25% of familial ALS and FTD, respectively, and approximately 6% of sporadic ALS (Majounie *et al*, 2012). C9ALS/FTD cases are distinguished from other ALS and FTD cases by specific ubiquitin and p62-positive but TDP-43-negative, neuronal cytoplasmic and intranuclear inclusions in the cerebellum and hippocampus (Al-Sarraj *et al*, 2011; Cooper-Knock *et al*, 2012; Mackenzie *et al*, 2014).

Reduced levels of *C9orf72* mRNA have been reported in post-mortem tissue, patient-derived lymphoblast cell lines and iPSNs and in blood samples (DeJesus-Hernandez *et al*, 2011; Cooper-Knock *et al*, 2012; Gijselinck *et al*, 2012, 2015; Belzil *et al*, 2013; Ciura *et al*, 2013; Donnelly *et al*, 2013; Mori *et al*, 2013b; Xi *et al*, 2013) and reduced C9orf72 protein levels were detected in the frontal cortex of ALS and FTD cases (Waite *et al*, 2014; Xiao *et al*, 2015), suggesting haploinsufficiency may contribute to C9ALS/FTD by a loss-of-function mechanism. In addition, the pathogenic mechanism may involve a toxic gain of function via RNA toxicity (DeJesus-Hernandez *et al*, 2011; Donnelly *et al*, 2013; Lagier-Tourenne *et al*, 2013; Mizielinska *et al*, 2013; Mori *et al*, 2013b; Sareen *et al*, 2013; Zu *et al*, 2013) and/or repeat-associated non-ATG (RAN) translation of the repeats into aggregation-prone dipeptide-repeat (DPR) proteins (Ash *et al*, 2013; Gendron *et al*, 2013; Mori *et al*, 2013a,b;

1   Sheffield Institute for Translational Neuroscience (SITraN), Department of Neuroscience, University of Sheffield, Sheffield, UK
2   Department of Biomedical Science, University of Sheffield, Sheffield, UK
3   The Research Institute at Nationwide Children's Hospital, Columbus, OH, USA
    *Corresponding author. Tel: +44 114 222 2241; E-mail: k.de_vos@sheffield.ac.uk
    ‡These authors contributed equally to this work
    §These authors contributed equally to this work
    †Present address: MRC Mitochondrial Biology Unit, Cambridge, UK

Zu *et al*, 2013; Mizielinska *et al*, 2014). Possibly all three mechanisms contribute to the pathogenesis of C9ALS/FTD in a multiple-hit fashion.

Alternative splicing of *C9orf72* is predicted to yield three mRNAs that code for two C9orf72 isoforms, a 481 amino acid (aa) isoform of approximately 50 kDa, C9orf72L (aa 1–481) and a 222 aa, 25 kDa isoform, C9orf72S (aa 1–221), respectively (DeJesus-Hernandez *et al*, 2011). Data from *C9orf72* knockout mice have shown that C9orf72 is required for macrophage and microglial function *in vivo* (Atanasio *et al*, 2016; O'Rourke *et al*, 2016; Sudria-Lopez *et al*, 2016; Sullivan *et al*, 2016), but the neuronal function of C9orf72 remains unclear. It has been suggested that C9orf72 regulates endosomal/lysosomal trafficking via Rab GTPases, but details are lacking (Farg *et al*, 2014). Alternatively, C9orf72 may be involved in nucleocytoplasmic shuttling (Xiao *et al*, 2015). Bioinformatics analysis revealed that C9orf72 shows structural homology to the Differentially Expressed in Normal and Neoplasia (DENN) family of proteins which function as GDP/GTP exchange factors (GEF) of Rab GTPases (Zhang *et al*, 2012; Levine *et al*, 2013).

Here, we show that C9orf72 is a Rab1a effector that controls initiation of autophagy by regulating the Rab1a-dependent trafficking of the ULK1 autophagy initiation complex to the phagophore. Disruption of C9orf72 function in cell lines and primary neurons inhibited autophagy and caused p62 accumulation similar to the specific pathology observed in C9ALS/FTD patients. Furthermore, basal autophagy levels were reduced in C9ALS/FTD patient-derived iNeurons. Together, these findings identify a novel function of C9orf72 and suggest a possible role for autophagy deficits by C9orf72 haploinsufficiency in C9ALS/FTD.

## Results

### C9orf72 regulates the initiation of autophagy

C9ALS/FTD patients show specific ubiquitin and p62-positive but TDP-43-negative, neuronal cytoplasmic and intranuclear inclusions in the cerebellum and hippocampus, which is indicative of impaired autophagy (Al-Sarraj *et al*, 2011; Cooper-Knock *et al*, 2012; Mackenzie *et al*, 2014). Because C9orf72 haploinsufficiency may contribute to C9ALS/FTD by a loss-of-function mechanism, we hypothesized that C9orf72 may be involved in autophagy.

To test this directly, we reduced the expression of both C9orf72 isoforms in HeLa or HEK293 cells using a pool of two siRNAs directed against different regions of C9orf72, and in rat primary cortical neurons using miRNA and determined the effect of C9orf72 depletion on autophagy by monitoring microtubule-associated protein 1A/1B light chain 3 (LC3) flux (Spira *et al*, 1993). During autophagy, the cytosolic form of LC3 (LC3-I) is lipidated to form LC3-II, which is recruited to autophagosomal membranes. LC3-II remains membrane associated throughout autophagy, and in the process, it is degraded in autolysosomes. Thus, turnover of LC3-II reflects the progression of autophagy. Due to questionable specificity of commercial C9orf72 antibodies, we used mRNA levels and RT–qPCR to confirm C9orf72 knockdown throughout this study (Appendix Fig S2). In mammalian cells, ULK1 controls the initial stages of autophagosome formation and is itself regulated in response to nutrient starvation by mammalian target of rapamycin (mTOR) kinase. Therefore, inhibitors of mTOR such as rapamycin and Torin1 are reliable experimental inducers of ULK1-dependent autophagy (Thoreen *et al*, 2009).

HeLa cells treated with non-targeting control siRNA or C9orf72 siRNA were transfected with mCherry-EGFP-LC3, and autophagy was induced using Torin1. We quantified the number of autophagosomes and autolysosomes by counting mCherry-EGFP-LC3-positive puncta by fluorescence microscopy. mCherry-EGFP-LC3 allows distinction between autophagosomes and autolysosomes because EGFP fluorescence is quenched by the low pH in autolysosomes whereas mCherry fluorescence is not affected (Kimura *et al*, 2007; Pankiv *et al*, 2007). As expected, Torin1 treatment caused a large increase in autophagosomes and a more modest increase in autolysosomes in control siRNA-treated cells. By contrast, in C9orf72 siRNA-treated cells no increase in autophagosomes was observed after Torin1 treatment, and the increase in autolysosomes was significantly reduced (Fig 1A). Thus, C9orf72 siRNA appeared to interfere with the induction of autophagy but not the later stages. To directly investigate this possibility, we treated HeLa cells with bafilomycin A1, a V-ATPase inhibitor that causes an increase in lysosomal pH and blocks fusion of autophagosomes with lysosomes (Spira *et al*, 1993). Under these conditions, counting autophagosomes allows specific quantification of the induction of autophagy (Tanida *et al*, 2005). In control siRNA-treated HeLa cells incubated with bafilomycin A1, Torin1 treatment induced

---

**Figure 1.  C9orf72 regulates the initiation of autophagy.**

A, B  HeLa cells treated with non-targeting siRNA (Ctrl) or C9orf72 siRNA and transfected with mCherry-EGFP-LC3 were treated with vehicle (Ctrl), Torin1 (250 nM; 3 h), bafilomycin A1 (BafA1, 100 nM; 6 h), or combinations thereof as indicated. Autophagosomes (green+red) and autolysosomes (red only) were quantified per cell (mean ± SEM from 3 independent experiments; one-way ANOVA with Fisher's LSD test: ns, not significant, *$P \leq 0.05$, **$P \leq 0.01$, ***$P \leq 0.001$, ****$P \leq 0.0001$; *N* (cells) = Ctrl/Ctrl: 120; Ctrl/Torin1: 101; C9orf72/Ctrl: 99; C9orf72/Torin1: 106; Ctrl/BafA1: 116; Ctrl/Torin1/BafA1: 118; C9orf72/BafA1: 109; C9orf72/Torin1/BafA1: 106). Scale bar = 20 μm. C9orf72 knockdown was confirmed by RT–qPCR (Appendix Fig S2).

C, D  HEK293 cells treated with non-targeting (Ctrl) or C9orf72 siRNA were incubated with BafA1, BafA1 + Torin1 (C), or BafA1 + rapamycin (D), and levels of LC3-I and II were determined by immunoblots. Levels of LC3-II were normalized against α-tubulin and are shown relative to the BafA1-treated sample (mean ± SEM; one-way ANOVA with Fisher's LSD test: ns, not significant, *$P \leq 0.05$, **$P \leq 0.01$, ***$P \leq 0.001$, ****$P \leq 0.0001$; *N* = 3 experiments).

E, F  Primary cortical neurons (DIV5/6) were transfected with ECFP non-targeting (Ctrl) or C9orf72 miRNA (cyan) and EGFP-LC3 (green). Three days post-transfection, neurons were treated with vehicle (Ctrl), Torin1 (250 nM; 3 h), BafA1 (100 nM; 5 h), or combinations thereof as indicated. Autophagosomes were quantified as the number of EGFP-LC3-positive puncta per soma from 2 independent experiments (mean ± SEM; one-way ANOVA with Fisher's LSD test: ns, not significant, *$P \leq 0.05$; ***$P \leq 0.001$; ****$P \leq 0.0001$; *N* (cells) = Ctrl miRNA/Ctrl: 77; Ctrl miRNA/Torin1: 68; C9orf72 miRNA/Ctrl: 57; C9orf72 miRNA/Torin1: 69; Ctrl miRNA/BafA1: 35; Ctrl miRNA/Torin1/BafA1: 57; C9orf72 miRNA/BafA1: 66; C9orf72 miRNA/Torin1/BafA1: 64). Scale bar = 5 μm.

Source data are available online for this figure.

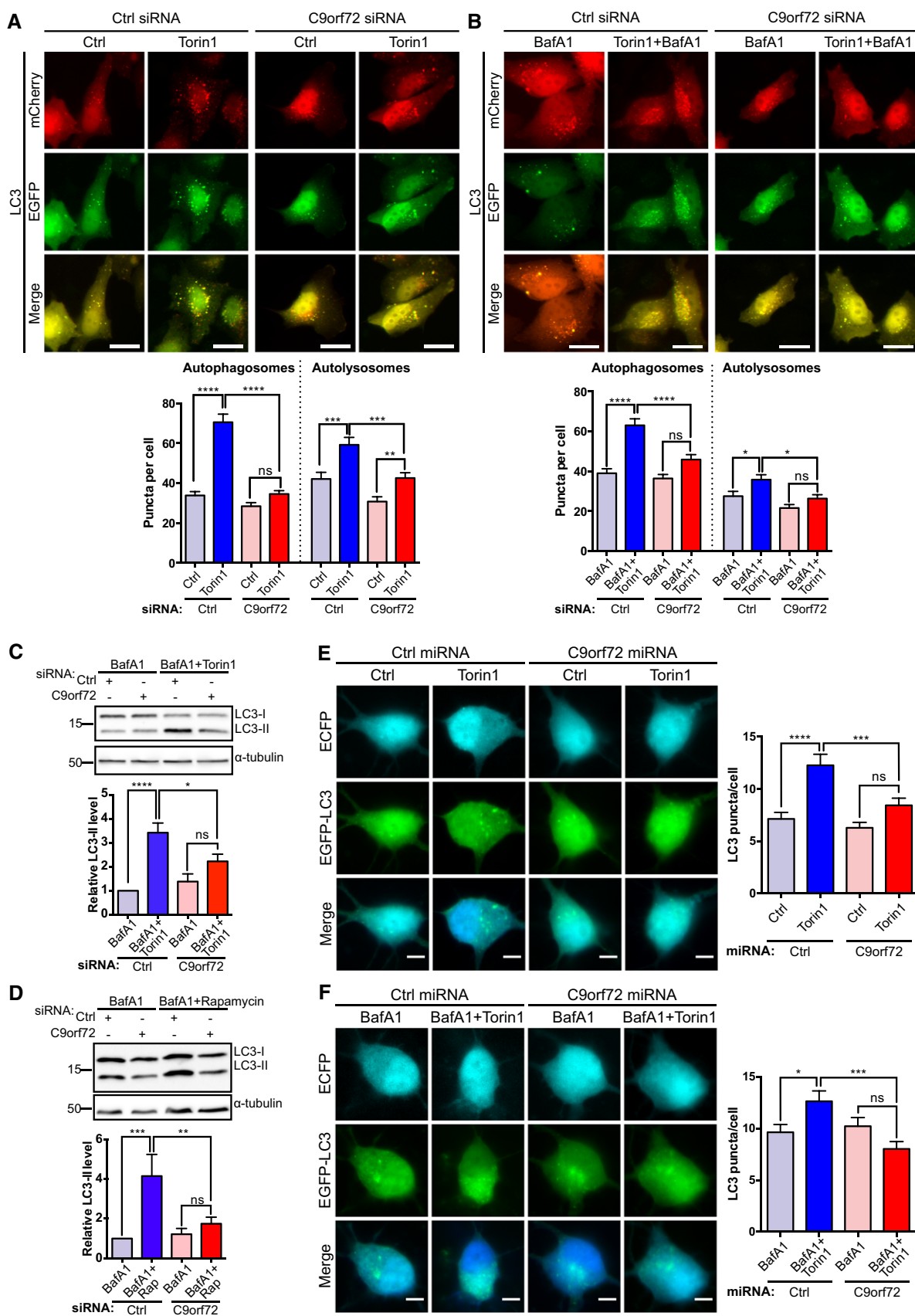

**Figure 1.**

autophagy and caused an increase in autophagosomes proportional to the level of induction. Conversely, in C9orf72 siRNA-treated HeLa cells no increase was detected (Fig 1B), indicating that loss of C9orf72 prevents induction of autophagy.

In a complementary approach, we quantified the levels of endogenous LC3-II on immunoblots. In HEK293 cells treated with non-targeting control siRNA, inhibition of mTOR with either Torin1 (Figs 1C and EV1A) or rapamycin (Figs 1D and EV1B) in the presence of bafilomycin A1 caused a marked increase in LC3-II levels compared to control cells. However, consistent with inhibition of autophagy initiation, LC3-II levels did not increase upon induction of autophagy in HEK293 cells treated with C9orf72 siRNA (Fig 1C and D), confirming the results obtained with mCherry-EGFP-LC3 in HeLa cells. To confirm the specificity of the siRNAs, we also tested the single siRNAs separately in this assay and found that both siRNAs knocked down C9orf72 and inhibited initiation of autophagy (Appendix Fig S1).

To assess whether loss of C9orf72 also affected induction of autophagy in neurons, we co-transfected primary rat cortical neurons with EGFP-LC3 and non-targeting control miRNA or C9orf72 targeting miRNA and quantified the number of EGFP-LC3-positive autophagosomes after treatment with Torin1 and/or bafilomycin A1. Compared to non-treated controls, Torin1 treatment caused an increase in autophagosomes in control miRNA-transfected neurons but not in C9orf72 miRNA-transfected neurons (Fig 1E). Similarly, in control miRNA expressing neurons incubated with bafilomycin A1, Torin1 treatment induced autophagy and caused an increase in autophagosomes, but this was not the case in C9orf72 miRNA-transfected neurons (Fig 1F). Hence, loss of C9orf72 also prevents induction of autophagy in primary neurons.

We next asked whether C9orf72 could induce autophagy and whether this involved the ULK1 initiation complex. FAK family kinase-interacting protein of 200 kDa (FIP200) forms a stable complex with ULK1 and autophagy-related 13 (ATG13) in the ULK1 initiation complex and is essential for the initiation of autophagy. Hence, FIP200 siRNA can be used to disrupt the ULK1 initiation complex (Hara *et al*, 2008). We overexpressed C9orf72 in cells treated with non-targeting or FIP200 siRNA and monitored induction of autophagy by determining co-transfected EGFP-LC3 conversion on immunoblots and by investigating the distribution of EGFP-LC3. In control siRNA-treated cells, overexpression of either C9orf72S or C9orf72L increased EGFP-LC3-II levels (Fig 2A) and correspondingly the number of EGFP-LC3-positive autophagosomes increased (Fig 2B). Loss of FIP200 completely prevented induction of autophagy by expression of C9orf72 in both assays (Figs 2A and B). As expected, FIP200 siRNA also completely prevented induction of autophagy by Torin1 treatment (Fig 2B). Thus, C9orf72 overexpression induces autophagy via the ULK1 complex.

Together, these results reveal that C9orf72 regulates autophagy initiation in cell lines and primary neurons via the ULK1 complex.

### C9orf72 interacts with the ULK1 initiation complex

To further characterize the involvement of C9orf72 in autophagy initiation, we investigated the interaction of C9orf72 with the ULK1 complex in co-immunoprecipitation assays. In a first approach, HEK293 cells were transfected with Myc- or FLAG-tagged C9orf72S or C9orf72L together with FLAG-FIP200, HA-ULK1, or Myc-ATG13

and C9orf72 was isolated using antibodies against the Myc or FLAG tag. FLAG-FIP200, HA-ULK1, and Myc-ATG13 specifically co-immunoprecipitated with C9orf72S and C9orf72L in these assays (Figs 3A–C). Secondly, we immunoprecipitated transfected Myc-tagged C9orf72S or C9orf72L and probed the resulting immune pellet for endogenous FIP200, ULK1, and ATG13. All three endogenous members of the ULK1 initiation complex were detected in the C9orf72 immunoprecipitates (Fig 3D and E).

We next investigated the interaction of C9orf72 with the ULK1 initiation complex *in situ* by proximity ligation assay (PLA) (Söderberg *et al*, 2006) in HeLa cells transfected with HA-ULK1 or FLAG-FIP200 and Myc-C9orf72 or with Myc-ATG13 and EGFP-C9orf72. Using antibodies against the tags, we observed proximity signals in all co-transfected cells examined (Figs 3F–H and EV2), confirming the interaction of C9orf72 with the ULK1 initiation complex in intact cells. As negative controls, we transfected HeLa cells with only C9orf72, ULK1, FIP200, or ATG13 and found only very low numbers of proximity signals.

To further characterize the interaction of C9orf72 with the ULK1 complex, we tested for interactions in *in vitro* binding assays. We incubated recombinant GST-tagged C9orf72 with *in vitro*-translated $^{35}$S-radiolabeled FIP200-6xHis segments (aa 1–638, aa 639–1373, and aa 1374–1591), HA-ULK1 or Myc-ATG13 and pulled down GST-C9orf72 using glutathione beads. C9orf72S and C9orf72L bound to the N-terminus of FIP200 (aa 1–638) but not the middle segment aa 639–1373 (Fig 3I). The C-terminus of FIP200 (aa 1374–1591) bound to C9orf72S and C9orf72L but also to the GST control, indicating that this was not a specific interaction (Fig 3I). C9orf72S and C9orf72L also interacted with ATG13 and ULK1 (Fig 3J and K).

Together, these data show that C9orf72 interacts with the ULK1 initiation complex by binding to ULK1, ATG13, and the N-terminus of FIP200.

### Loss of C9orf72 does not affect ULK1 activation

Under basal conditions, mTOR suppresses ULK1 activity by phosphorylating ULK1 at Ser757 (Kim *et al*, 2011). Inactivated mTOR is released from the ULK1 complex and ULK1 Ser757 phosphorylation is lost, thereby enhancing ULK1 kinase activity, which in turn phosphorylates ATG13 and FIP200 to initiate autophagy (Ganley *et al*, 2009; Jung *et al*, 2009; Kim *et al*, 2011). Thus, Ser757 phosphorylation of ULK1 is indicative of the activation state of the ULK1 complex. To investigate whether C9orf72 affected autophagy at the level of ULK1 activation, we monitored ULK1 Ser757 phosphorylation using phosphospecific antibodies. Inhibition of mTOR by rapamycin reduced phosphorylation of ULK1 on Ser757 in both control and C9orf72 siRNA-treated HEK293 cells (Fig 4A). Thus, loss of C9orf72 did not prevent activation of ULK1.

### C9orf72 regulates translocation of the ULK1 complex to the phagophore via Rab1a

The activated ULK1 complex translocates to the phagophore to initiate autophagosome formation (Hara *et al*, 2008). To investigate whether C9orf72 was involved in the translocation of the ULK1 complex to the phagophore, we monitored the translocation of mCherry-FIP200 in HeLa cells and primary rat cortical neurons treated with siRNA or transfected with miRNA to knock down

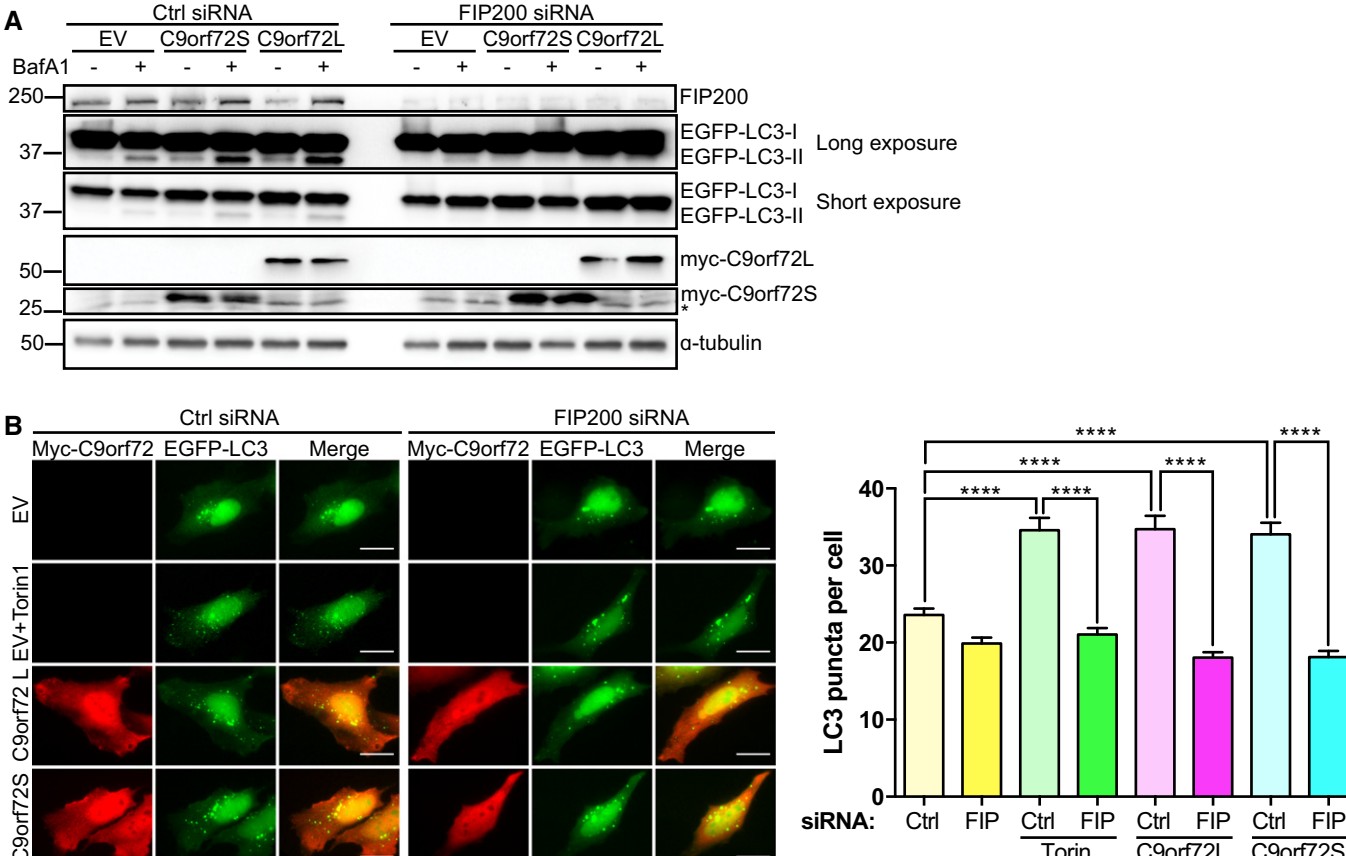

**Figure 2.   C9orf72 induces autophagy via the ULK1 complex.**

A   HEK293 cells treated with non-targeting (Ctrl) or FIP200 siRNA were co-transfected with EGFP-LC3 and either empty vector control (EV), Myc-C9orf72S, or Myc-C9orf72L. 24 h post-transfection cells were treated with vehicle or 100 nM BafA1 for 6 h. Samples were lysed and subjected to SDS–PAGE and immunoblot. Autophagy levels were determined by immunoblot for EGFP-LC3-I and II. Expression of Myc-C9orf72 was confirmed using anti-Myc (* indicates a nonspecific band). FIP200 knockdown was confirmed using anti-FIP200 antibodies. α-tubulin was used as a loading control.

B   HeLa cells treated with non-targeting (Ctrl) or FIP200 siRNA were co-transfected with empty vector (EV), Myc-C9orf72S or Myc-C9orf72L (red), and EGFP-LC3 (green) to label autophagosomes. As positive control, EV-transfected cells were treated for 3 h with Torin1 (250 nM). Autophagy was quantified as the number of EGFP-LC3-positive autophagosomes per cell from 3 independent experiments (mean ± SEM; one-way ANOVA with Fisher's LSD test: ****$P \leq 0.0001$, $N$ (cells) = Ctrl/EV: 73; FIP200/EV: 76; Ctrl/EV/Torin1: 73, FIP200/EV/Torin1: 70; Ctrl/C9orf72L: 71; FIP200/C9orf72L: 74; Ctrl/C9orf72S: 72; FIP200/C9orf72S: 72). Scale bar = 20 μm. FIP200 knockdown was confirmed by immunoblot (Appendix Fig S2).

Source data are available online for this figure.

C9orf72 expression, respectively. Under basal conditions, mCherry-FIP200 was found diffusely in the cytoplasm with some bright puncta observable in both control and C9orf72 knockdown HeLa cells (Fig 4B) and rat cortical neurons (Fig 4C). When we induced autophagy with Torin1, the number of mCherry-FIP200 puncta increased significantly in control cells, consistent with translocation of the ULK1 complex to the phagophore. In contrast, in C9orf72 knockdown cells and neurons, no increase above basal level was observed after treatment with Torin1 (Fig 4B and C). To further assert the specificity of this effect, we reintroduced C9orf72 into C9orf72 miRNA-transfected rat cortical neurons by transfection with human C9orf72. In these neurons, Torin1 treatment increased the number of mCherry-FIP200 puncta, showing that human C9orf72 rescued the inhibition of ULK1 complex translocation (Fig 4C). Interestingly, overexpression of human C9orf72 per se significantly increased the number of mCherry-FIP200 puncta, indicating that

increasing C9orf72 levels drives translocation of the ULK1 complex in absence of autophagy activators (Fig 4C). To test this further, we overexpressed C9orf72 in HeLa cells and monitored mCherry-FIP200 translocation. Overexpression of either C9orf72 isoform significantly increased the number of mCherry-FIP200 puncta to similar levels as found after Torin1 treatment (Fig 4D). Thus, C9orf72 is required for translocation of the ULK1 complex in cell lines and primary neurons.

In yeast, Ypt1 (the yeast homolog of Rab1) regulates translocation of Atg1 (ULK1 in mammals) to the phagophore (Lynch-Day *et al*, 2010; Wang *et al*, 2013). To determine whether this was also the case in mammalian cells, we investigated mCherry-FIP200 translocation in HeLa cells and SH-SY5Y neuroblastoma cells in which we depleted Rab1a by siRNA—we chose Rab1a because it had been shown to regulate autophagy in mammalian cells (Winslow *et al*, 2010; Mukhopadhyay *et al*, 2011) and because we

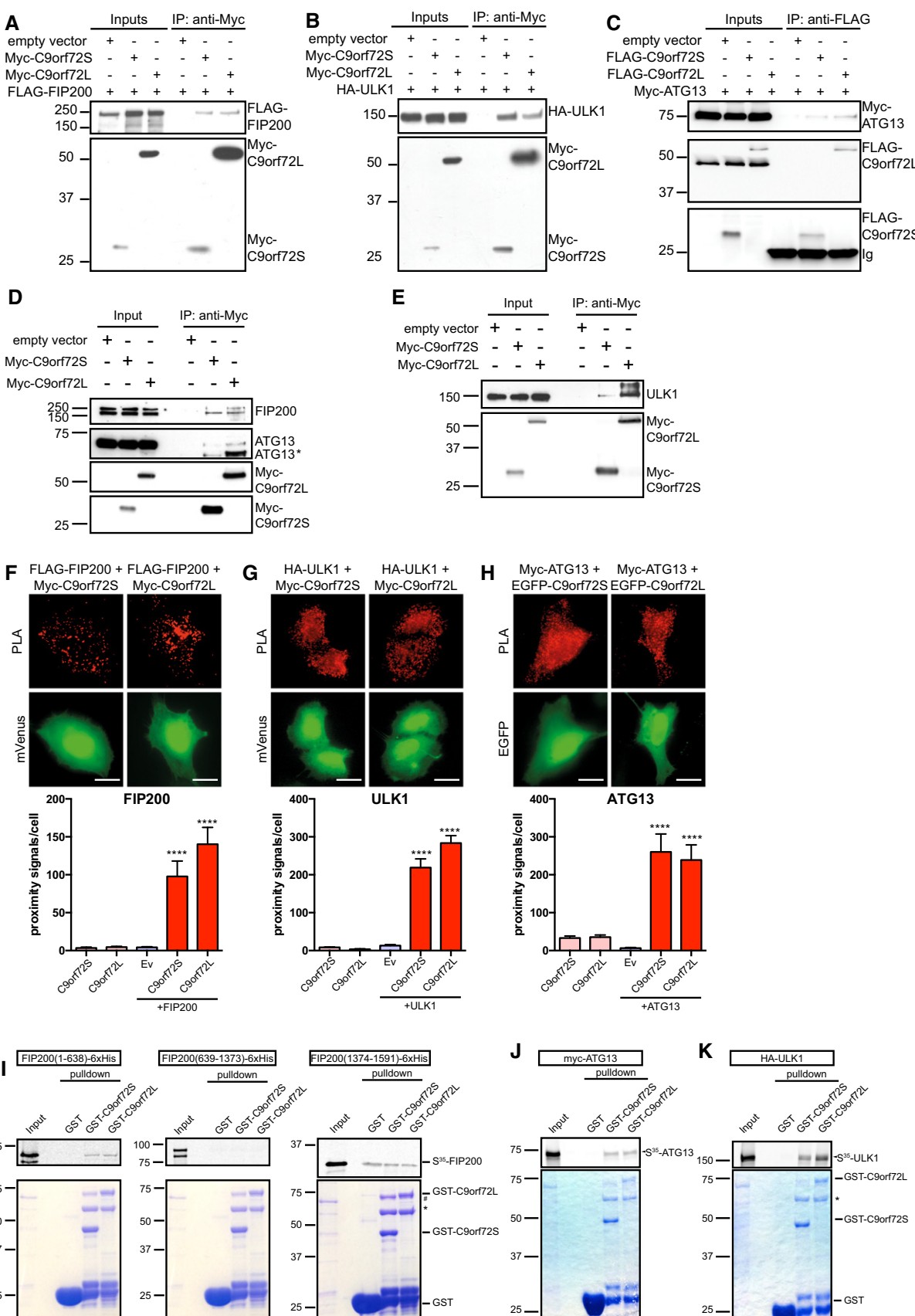

**Figure 3.**

**Figure 3.  C9orf72 interacts with the ULK1 initiation complex.**

A–C  Cell lysates of HEK293 cells co-transfected with FLAG-FIP200 (A), or HA-ULK1 (B) and either empty vector control, Myc-C9orf72S, or Myc-C9orf72L or with Myc-ATG13 (C) and either empty vector control, FLAG-C9orf72S, or FLAG-C9orf72L were subjected to immunoprecipitation with anti-Myc (A and B) or anti-FLAG (C) antibodies. Immune pellets were probed for Myc-C9orf72 (A and B), FLAG-C9orf72 (C), FLAG-FIP200 (A), HA-ULK1 (B), or Myc-ATG13 (C) on immunoblots.

D, E  Cell lysates of HEK293 cells transfected with empty vector control, Myc-C9orf72S, or Myc-C9orf72L were subjected to immunoprecipitation with anti-Myc antibodies. Immune pellets were probed for Myc-C9orf72 and endogenous FIP200, ULK1, and ATG13. There are multiple alternatively spliced forms of ATG13 (Jung et al, 2009; Alers et al, 2011); ATG13* is most likely a smaller alternative spliced form of ATG13 that is enriched by interaction with C9orf72.

F–H  HeLa cells transfected with HA-ULK1 or Flag-FIP200 and Myc-C9orf72 or with Myc-ATG13 and EGFP-C9orf72 were fixed and processed for PLA analysis. Transfections were laced with mVenus to enable identification of transfected cells for analysis where required (green). PLA signals (red) were counted per cell (mean ± SEM; one-way ANOVA with Fisher's LSD test, ****$P \leq 0.0001$; N (cells) = (F) C9orf72S: 22; C9orf72L: 18; EV+FIP200: 18; C9orf72S+FIP200: 18; C9orf72L+FIP200: 17; (G) C9orf72S: 21; C9orf72L: 20; EV+ULK1: 20; C9orf72S+ULK1: 22; C9orf72L+ULK1: 20; (H) C9orf72S: 11; C9orf72L: 10; EV+ATG13: 11; C9orf72S+ATG13: 11; C9orf72L+ATG13: 11). Scale bar = 10 μm; see also Fig EV2.

I–K  $^{35}$S-radiolabeled recombinant FIP200-6xHis segments (I), Myc-ATG13 (J), or HA-ULK1 (K) were added to GST, GST-C9orf72S, and GST-C9orf72L immobilized on glutathione-coated beads. $^{35}$S-radiolabeled recombinant proteins were visualized by phosphorimager (top panels). Coomassie-stained GST, GST-C9orf72S, and GST-C9orf72L in the pull-down samples are shown (bottom panels). The identity of the Coomassie protein bands was confirmed by mass spectrometry ([#] indicates E. coli DnaK chaperonin; * indicates E. coli 60kD chaperonin; Appendix Fig S3).

Source data are available online for this figure.

had identified Rab1a as a potential binding partner of C9orf72 by yeast 2 hybrid (Y2H, Fig 6A). Thus, our data suggested that C9orf72 might act with Rab1a to regulate translocation of the ULK1 complex and induce autophagy. To investigate this possibility, we quantified the induction of ULK1 complex translocation and autophagy elicited by increased C9orf72S or C9orf72L expression in Rab1a siRNA-treated HeLa and SH-SY5Y neuroblastoma cells. Rab1a siRNA completely prevented translocation of mCherry-FIP200 induced by Torin1, confirming its role in ULK1 complex trafficking (Fig 5A and B). Overexpression of C9orf72S or C9orf72L no longer induced translocation of mCherry-FIP200 in Rab1a siRNA-treated cells, and the number of C9orf72-induced EGFP-LC3-positive autophagosomes was significantly reduced (Fig 5A–C). In an alternative approach, we used dominant negative (DN) Rab1a to inhibit Rab1a in HeLa cells expressing Myc-C9orf72S or C9orf72L and monitored mCherry-FIP200 and EGFP-LC3. Again inhibition of Rab1a function reduced mCherry-FIP200 translocation and autophagosome formation elicited by increased C9orf72S or C9orf72L expression (Fig EV3). Thus, C9orf72 regulates Rab1a-dependent trafficking of the ULK1 complex.

## C9orf72 interacts with Rab1a

Our data and the structural homology of C9orf72 with DENN domain proteins that are GEFs for Rab GTPases (Zhang et al, 2012; Levine et al, 2013) suggested that C9orf72 might interact with Rab1a. Consistent with a direct interaction, we isolated a partial Rab1a clone (aa 46–205) in a Y2H screen of a human brain cDNA library with C9orf72S as "bait" (Fig 6A). To test this interaction further, we first transfected HEK293 cells with Myc-tagged Rab1a and FLAG-tagged C9orf72S or C9orf72L and performed co-immunoprecipitation assays. Secondly, we immunoprecipitated transfected Myc-tagged C9orf72S or C9orf72L and probed the resulting immune pellet for endogenous Rab1a. In both assays, Rab1a co-immunoprecipitated efficiently with C9orf72L and to a lesser extent with C9orf72S (Fig 6B and C).

We next asked whether C9orf72 interacted with Rab1a in in vitro binding assays. We incubated recombinant GST-tagged C9orf72 with in vitro-translated $^{35}$S-radiolabeled Rab1a and pulled down GST-C9orf72 using glutathione beads. Rab1a interacted with both C9orf72L and C9orf72S in these assays (Fig 6D). Preloading

Rab1a with GDP to mimic inactive Rab1a did not alter binding to C9orf72, but preloading Rab1a with the non-hydrolysable GTP analog GMP-PNP to mimic activated Rab1a increased interaction of Rab1a with C9orf72S and C9orf72L two- and fourfold, respectively (Fig 6D). To further validate the specificity of the C9orf72–Rab1a interaction, we performed an equilibrium binding experiment using increasing concentrations of radiolabeled C9orf72 protein and a constant amount of GST-Rab1a. There was no binding to the GST-negative pull-down control and increased binding of C9orf72 to GST-Rab1a with increasing concentrations of C9orf72 protein added. Consistent with a specific interaction, the resulting binding curve fitted a hyperbola ($R^2 = 0.94$) which is characteristic of a biological interaction (Fig 6E; Pollard, 2010). Thus, C9orf72 binds to Rab1a and behaves as a Rab1a effector rather than a Rab1a GEF.

## C9orf72 mediates interaction of Rab1a with the ULK1 complex

We reasoned that C9orf72 may recruit activated Rab1a to the ULK1 complex to initiate translocation. To test this, we probed the interaction of HA-ULK1 and Myc-Rab1a in HeLa cells treated with non-targeting control siRNA or C9orf72 siRNA by in situ PLA. In control siRNA-treated cells, we observed proximity signals in all cells co-transfected with HA-ULK1 and Myc-Rab1a while in cells transfected with HA-ULK1 or Myc-Rab1a alone only very low numbers of proximity signals were observed. In contrast, in cells treated with C9orf72 siRNA proximity signals were no longer detected in HA-ULK1 and Myc-Rab1a-co-transfected cells (Fig 7A). Thus, these data show ULK1–Rab1a interaction in intact cells and reveal that this interaction is C9orf72 dependent.

To test the functional significance of these interactions, we transfected HeLa cells that were treated with control or C9orf72 siRNA with dominant active Rab1a(Q70L) and monitored mCherry-FIP200 translocation and EGFP-LC3-positive autophagosomes. Consistent with the Rab1a dependency of ULK1 complex translocation, in control siRNA-treated cells Rab1a(Q70L) induced translocation of mCherry-FIP200 and Rab1a(Q70L) appeared punctate and co-localized with FIP200-positive structures. In contrast, in C9orf72 siRNA-treated cells Rab1a(Q70L) appeared more uniformly distributed in the cytoplasm and did not induce FIP200 translocation (Fig 7B). Similarly, overexpression of Rab1a(Q70L) increased the

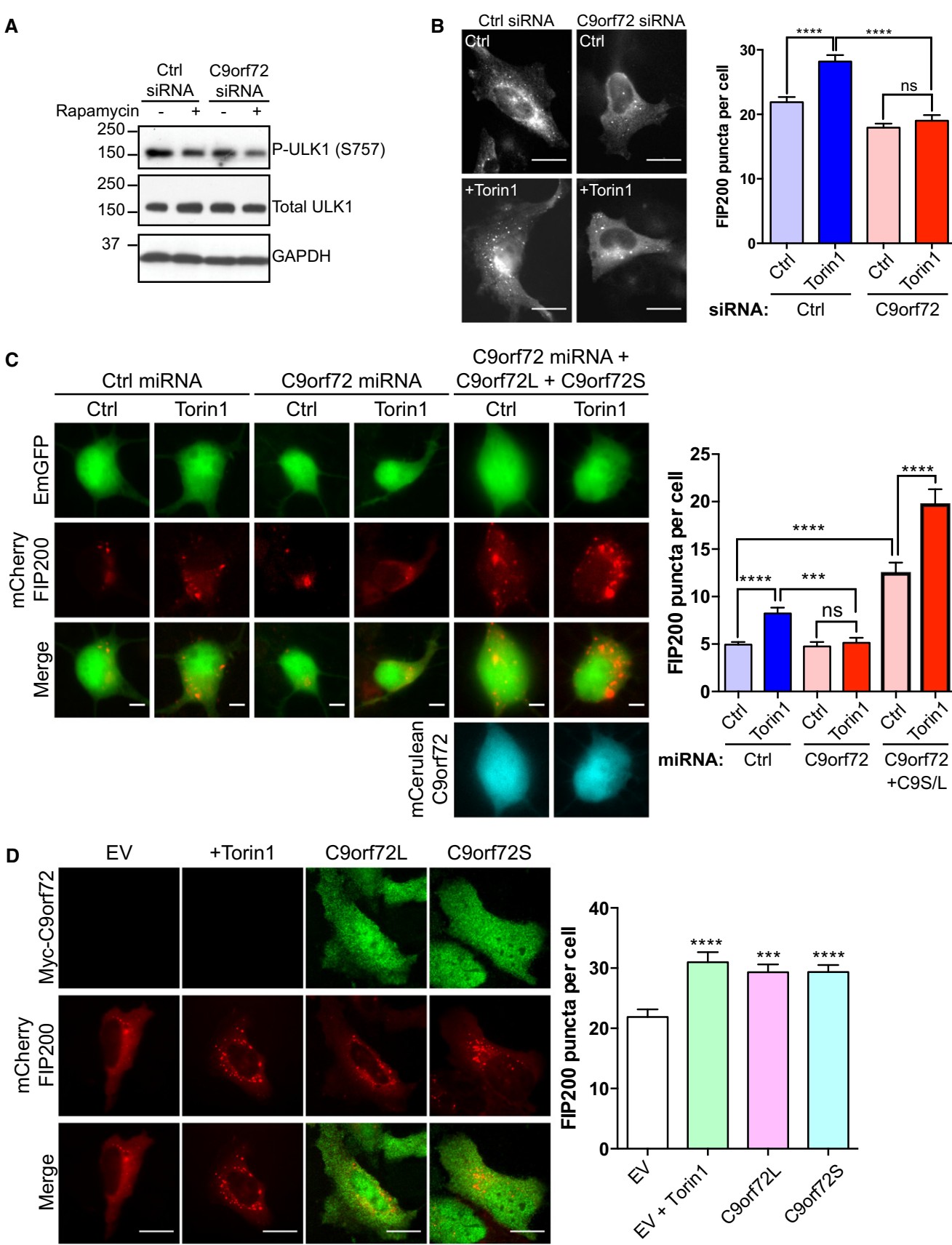

Figure 4.

**Figure 4.   C9orf72 regulates translocation of the ULK1 complex.**

A  HEK293 cells were transfected with non-targeting (Ctrl) or C9orf72 siRNA. Cells were treated with rapamycin for 6 h to induce autophagy. Activation of ULK1 was determined on immunoblots using phospho-ULK1 (Ser757), total ULK1, and GAPDH Abs (loading control).

B  HeLa cells treated with non-targeting (Ctrl) or C9orf72 siRNA were transfected with mCherry-FIP200. Twenty-four hours post-transfection, cells were treated for 3 h with Torin1 (250 nM) or vehicle (Ctrl). Translocation of the ULK1 complex was quantified as the number of mCherry-FIP200-positive puncta per cell from 3 independent experiments (mean ± SEM; one-way ANOVA with Fisher's LSD test, ns: not significant, ****$P \leq 0.0001$; $N$ (cells) = Ctrl/Ctrl: 65; Ctrl/Torin1: 60; C9orf72/Ctrl: 54; C9orf72/Torin1: 49). C9orf72 knockdown was determined by RT–qPCR (Appendix Fig S2). Scale bar = 10 μm.

C  Primary cortical neurons (DIV5/6) were transfected with EmGFP non-targeting (Ctrl) or C9orf72 miRNA (green) and mCherry-FIP200 (red); for rescue experiments, the cells were additionally transfected with mCerulean-tagged C9orf72s and C9orf72L (cyan). Three days post-transfection, neurons were treated for 3 h with Torin1 (250 nM) or vehicle (Ctrl). Translocation of the ULK1 complex was quantified as the number of mCherry-FIP200-positive puncta per soma from 2 independent experiments (mean ± SEM; one-way ANOVA with Fisher's LSD test, ns: not significant, ***$P \leq 0.001$, ****$P \leq 0.0001$; $N$ (cells) = Ctrl miRNA/Ctrl: 134; Ctrl miRNA/Torin1: 125; C9orf72 miRNA/Ctrl: 101; C9orf72 miRNA/Torin1: 78; C9orf72 miRNA+C9orf72L+C9orf72S: 41; C9orf72 miRNA+C9orf72L+C9orf72S/Torin1: 39). Scale bar = 5 μm.

D  HeLa cells were co-transfected with mCherry-FIP200 (red) and empty vector (EV), FLAG-C9orf72L, or FLAG-C9orf72S (green). As positive control, EV-transfected cells were treated for 3 h with Torin1 (250 nM). Translocation of the ULK1 complex was quantified as the number of mCherry-FIP200-positive puncta per cell from 3 independent experiments (mean ± SEM; one-way ANOVA with Fisher's LSD test, ***$P \leq 0.001$, ****$P \leq 0.0001$; $N$ (cells) = EV: 47, EV+Torin1: 31, C9orf72L: 46, C9orf72S: 45). Scale bar = 10 μm.

Source data are available online for this figure.

number of autophagosomes in control siRNA-treated cells but not in C9orf72 siRNA-treated cells (Fig 7C).

In conclusion, these data are consistent with a model in which C9orf72 acts as an effector of Rab1a that recruits active Rab1a to the ULK1 complex to promote translocation of the ULK1 complex to the phagophore during autophagy initiation.

**C9orf72 depletion induces p62 accumulation**

C9ALS/FTD appears to involve haploinsufficiency of C9orf72 (DeJesus-Hernandez *et al*, 2011; Cooper-Knock *et al*, 2012; Gijselinck *et al*, 2012, 2015; Belzil *et al*, 2013; Ciura *et al*, 2013; Donnelly *et al*, 2013; Mori *et al*, 2013b; Xi *et al*, 2013; Waite *et al*, 2014). Our data predict that C9orf72 haploinsufficiency would impair autophagy. In agreement with this, C9ALS/FTD patients show characteristic p62-positive, neuronal cytoplasmic and intranuclear inclusions in the cerebellum and hippocampus (Al-Sarraj *et al*, 2011; Cooper-Knock *et al*, 2012; Mackenzie *et al*, 2014). To further investigate whether loss of C9orf72 is sufficient to cause p62 alterations consistent with the pathology in patients, we monitored endogenous p62 by immunofluorescence in C9orf72 siRNA-treated HeLa cells and in primary cortical neurons transduced with C9orf72 miRNA. p62-positive puncta accumulated in both HeLa and primary neurons depleted of C9orf72 when compared to non-targeting control (Fig 8A and B). This accumulation was directly caused by loss of C9orf72 because reintroducing C9orf72 by transduction with human C9orf72 reduced the number of p62 puncta back to control level in C9orf72 miRNA knockdown rat cortical neurons (Fig 8B). Thus, loss of C9orf72 mimics C9ALS/FTD p62 pathology *in vitro*.

**Basal levels of autophagy are reduced in C9ALS/FTD patient-derived iNeurons**

To further test the possible involvement of C9orf72 haploinsufficiency in C9ALS/FTD, we analyzed basal levels of autophagy in C9ALS/FTD patient-derived iNeurons. iNeurons were obtained by differentiation of iNPCs of two C9ALS/FTD patients and age- and gender-matched controls (Meyer *et al*, 2014) (Appendix Table S1). As iNPCs are not clonally expanded, they do not display clonal variability. We confirmed the levels of C9orf72 in these cells by

RT–qPCR. Compared to the controls, all C9ALS/FTD cases showed a marked reduction in C9orf72, which is consistent with previous reports of haploinsufficiency in patient material (Appendix Fig S4). Both LC3-I and II were detectable in the controls and LC3-II accumulated upon treatment with bafilomycin A1 (Figs 9A and B). Untreated C9ALS/FTD cases displayed higher levels of LC3-I compared to their controls, but LC3-II levels were similar. Upon treatment with bafilomycin A1, LC3-II levels accumulated to detectable levels in the C9ALS/FTD cases, but the levels of LC3-II remained significantly below those found in bafilomycin A1-treated controls. Thus, while autophagy was not completely blocked, there was a marked reduction of autophagy in C9ALS/FTD iNeurons compared to control iNeurons (Fig 9A and B).

# Discussion

We show that C9orf72 interacts with Rab1a and the ULK1 complex and mediates interaction of Rab1a with the ULK1 initiation complex to facilitate its trafficking to the phagophore and initiate autophagosome formation. Our data are in contrast to a previous report indicating an increase in LC3-II in C9orf72 siRNA-treated SH-SY5Y cells which suggests a possible late autophagy defect (Farg *et al*, 2014). Our experiments in HEK293, HeLa, and SH-SY5Y cells and in primary rat cortical neurons did not yield any evidence for involvement of C9orf72 in the late steps of autophagy. All our data firmly point toward a role of C9orf72 in autophagy initiation: (i) autophagic flux assays using multiple inducers and readouts in multiple cell lines and primary neurons show effects consistent with initiation defects in C9orf72 knockdown cells (Fig 1), (ii) overexpression of C9orf72 induces autophagy in a FIP200-dependent way (Fig 2), (iii) C9orf72 interacts with FIP200, ULK1, and ATG13 in the ULK1 initiation complex *in vitro* and in cells (Fig 3), (iv) C9orf72 interacts with Rab1a in Y2H, *in vitro* and in cells to regulate translocation of the ULK1 complex which is known to be essential for autophagy initiation (Figs 4–7), and (v) our data in iNeurons give no indication for block in the later stages of autophagy (Fig 9).

*In silico* sequence analysis revealed that C9orf72 shows structural homology to DENN family proteins which function as GEFs for Rab GTPases (Zhang *et al*, 2012; Levine *et al*, 2013). Recent work has

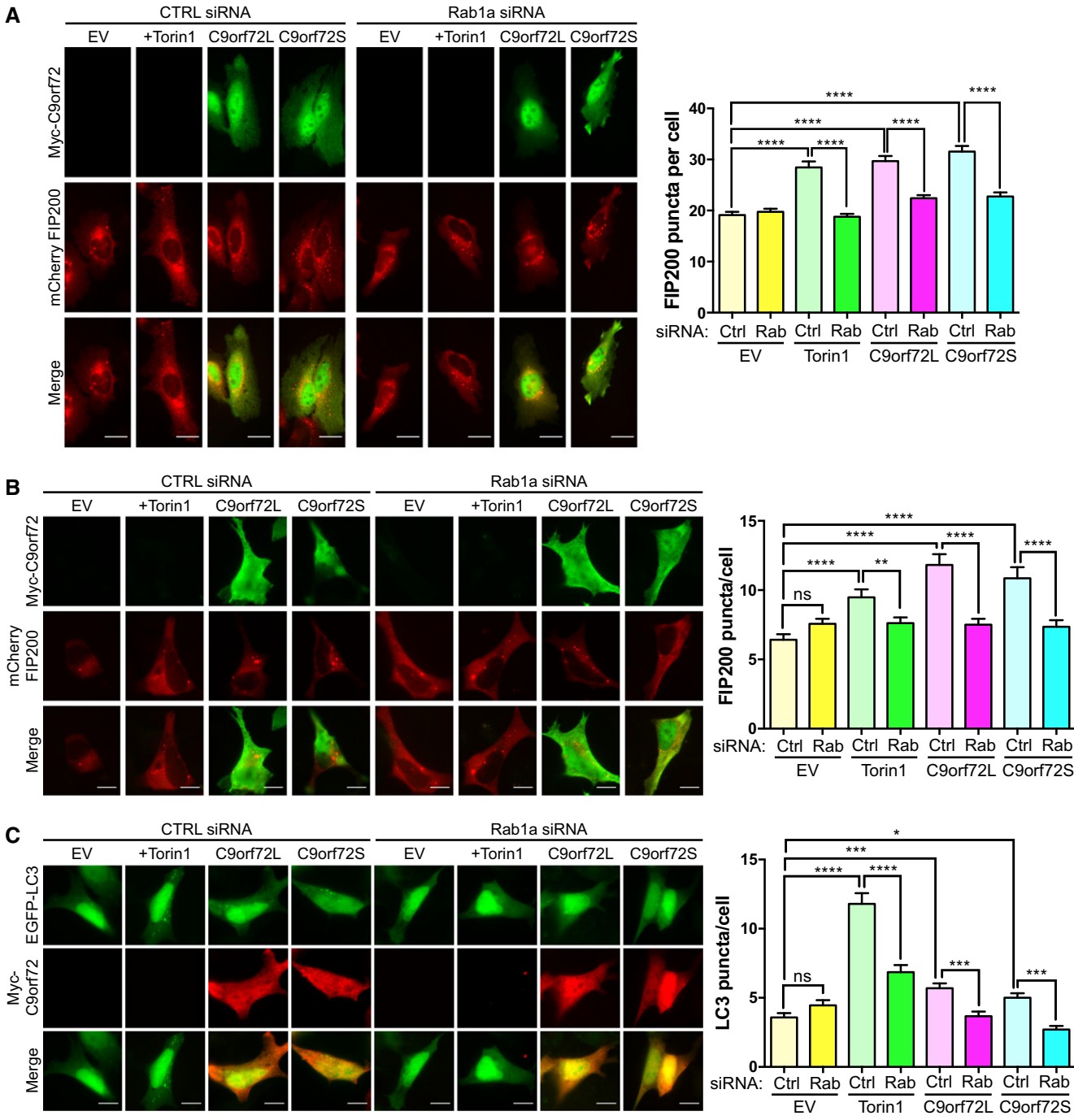

**Figure 5.  C9orf72 regulates translocation of the ULK1 complex via Rab1a.**

A, B   HeLa cells (A) or SH-SY5Y neuroblastoma cells (B) treated with non-targeting (Ctrl) or Rab1a siRNA were co-transfected with mCherry-FIP200 (red) and empty
       vector (EV), Myc-C9orf72L, or Myc-C9orf72S (green). As positive control, EV-transfected cells were treated for 3 h with Torin1 (250 nM). Translocation of the ULK1
       complex was quantified as the number of mCherry-FIP200-positive puncta per cell [A, HeLa, mean ± SEM; one-way ANOVA with Fisher's LSD test; ns, not
       significant; ****$P \leq 0.0001$; N (cells from 3 independent experiments) = Ctrl/EV: 81; Ctrl/EV/Torin1: 48; Ctrl/C9orf72L: 86; Ctrl/C9orf72S: 48; Rab1a/EV: 79; Rab1a/EV/
       Torin1: 68; Rab1a/C9orf72L: 78; Rab1a/C9orf72S: 52; B, SH-SY5Y: mean ± SEM; one-way ANOVA with Fisher's LSD test; ns, not significant; **$P \leq 0.01$; ****$P \leq 0.0001$;
       N (cells from 2 independent experiments) = Ctrl/EV: 70; Ctrl/EV/Torin1: 56; Ctrl/C9orf72L: 45; Ctrl/C9orf72S: 43; Rab1a/EV: 63; Rab1a/EV/Torin1: 55; Rab1a/C9orf72L:
       44; Rab1a/C9orf72S: 37]. Rab1a knockdown was confirmed by RT–qPCR (Appendix Fig S2). Scale bar = 10 μm.
C      SH-SY5Y neuroblastoma cells treated with non-targeting (Ctrl) or Rab1a siRNA were co-transfected with EGFP-LC3 (green) and empty vector (EV), Myc-C9orf72L, or
       Myc-C9orf72S (red). As positive control, EV-transfected cells were treated for 3 h with Torin1 (250 nM). Autophagosomes were quantified as the number of EGFP-
       LC3-positive puncta per cell (mean ± SEM; one-way ANOVA with Fisher's LSD test; ns, not significant; *$P \leq 0.05$, ***$P \leq 0.001$, ****$P \leq 0.0001$; N (cells from 2
       independent experiments) = Ctrl/EV: 102; Ctrl/EV/Torin1: 92; Ctrl/C9orf72L: 97; Ctrl/C9orf72S: 76; Rab1a/EV: 102; Rab1a/EV/Torin1: 107; Rab1a/C9orf72L: 101; Rab1a/
       C9orf72S: 87). Rab1a knockdown was confirmed by RT–qPCR (Appendix Fig S2). Scale bar = 10 μm.

                                                                                                 

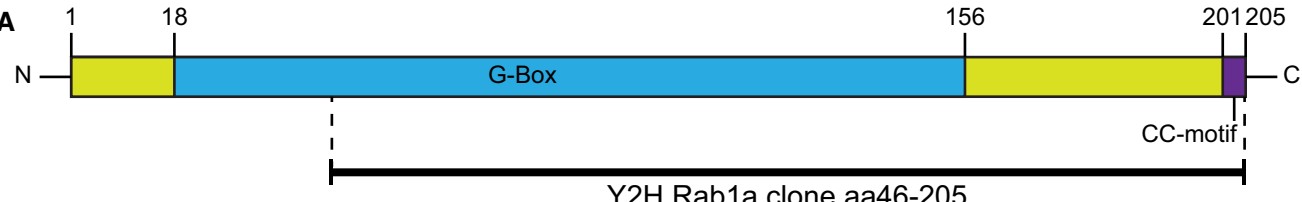

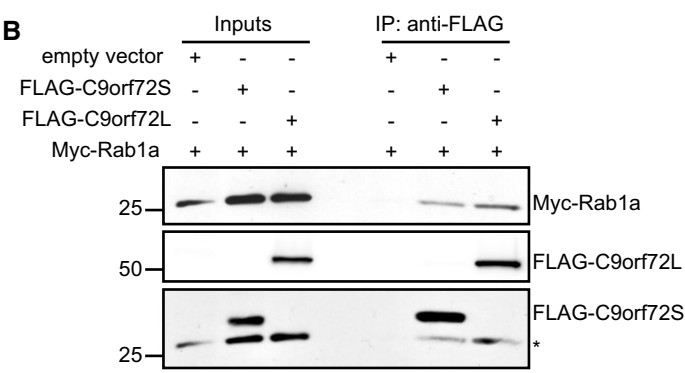

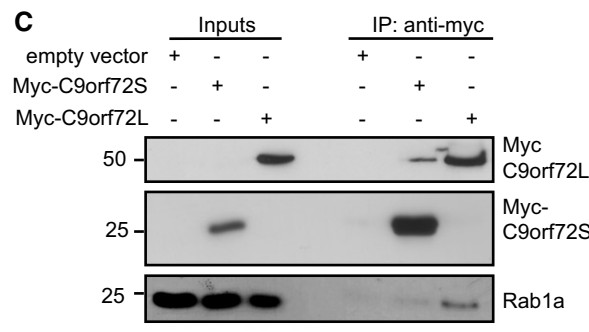

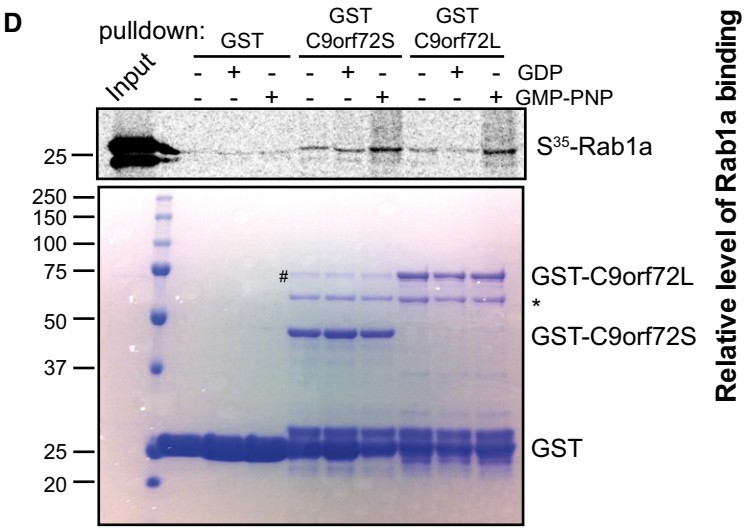

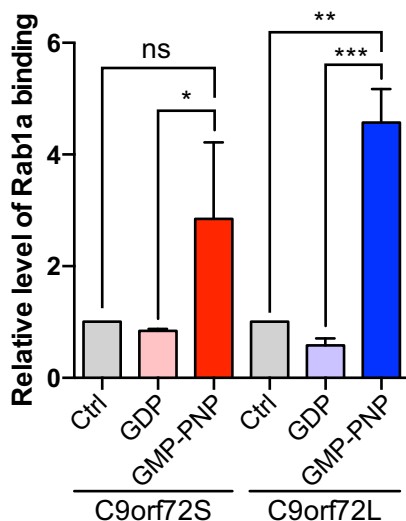

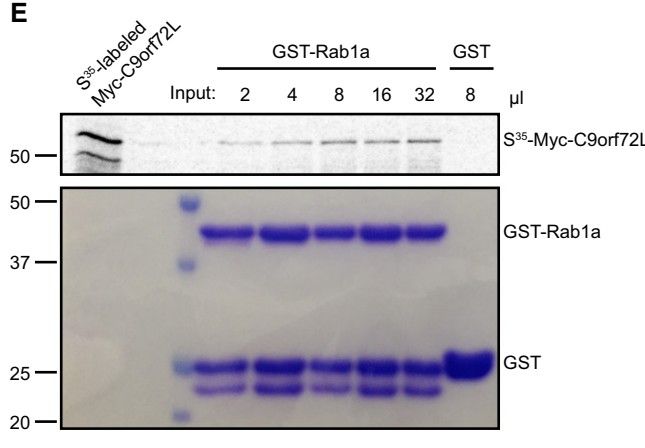

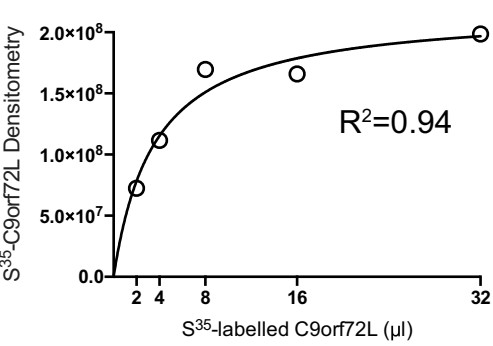

Figure 6.

**Figure 6.  C9orf72 interacts with Rab1a.**

A   A human brain random cDNA library was screened in a Y2H assay using C9orf72S as bait. A clone coding for aa 46–205 of Rab1a, comprising most of the GTP-binding G-box domain and the C-terminal CC-motif was found to interact with C9orf72S.

B   Cell lysates of HEK293 cells co-transfected with Myc-Rab1a and either empty vector, FLAG-C9orf72S, or FLAG-C9orf72L were subjected to immunoprecipitation with anti-FLAG antibody. Bound protein was eluted from beads using excess FLAG peptide. Immune eluates were probed for FLAG-C9orf72 and Myc-Rab1a on immunoblots. The input levels of FLAG-C9orf72 and Myc-Rab1a in the transfected cells are shown (Inputs). * indicates remaining Myc-Rab1a signal after reprobing for FLAG-C9orf72.

C   Cell lysates of HEK293 cells transfected with Myc-C9orf72S or Myc-C9orf72L were subjected to immunoprecipitation with anti-Myc antibody. The resulting immune pellet was probed for endogenous Rab1a.

D   $^{35}$S-radiolabeled recombinant Myc-Rab1a protein loaded with vehicle, GDP or GMP-PNP was added to GST, GST-C9orf72S, and GST-C9orf72L immobilized on glutathione-coated beads. $^{35}$S-radiolabeled recombinant Myc-Rab1a protein was visualized by phosphorimager (top panel). Coomassie-stained GST, GST-C9orf72S, and GST-C9orf72L in the pull-down samples are shown (bottom panel). The identity of the Coomassie protein bands was confirmed by mass spectrometry ($^{\#}$ indicates *E. coli* DnaK chaperonin; * indicates *E. coli* 60kD chaperonin; Appendix Fig S3). Relative binding of Rab1a to C9orf72 was quantified from 3 independent experiments (mean ± SEM; one-way ANOVA with Fisher's LSD test; ns, not significant; *$P \leq 0.05$; **$P \leq 0.01$; ***$P \leq 0.001$).

E   Increasing volumes of $^{35}$S-radiolabeled recombinant Myc-C9orf72L protein were incubated with equal amounts of GST-Rab1a immobilized on glutathione-coated beads in an equilibrium binding experiment. 8 µl of $^{35}$S-radiolabeled recombinant Myc-C9orf72L protein was incubated with GST as a negative control. Bound $^{35}$S-radiolabeled Myc-C9orf72L protein was visualized by phosphorimager. Coomassie-stained GST-Rab1a and GST in the pull-down samples are shown. Densitometry analysis of the amount of $^{35}$S-radiolabeled recombinant Myc-C9orf72L protein bound to GST-Rab1a in the different binding reactions was used to fit an equilibrium binding hyperbola ($R^2 = 0.94$).

Source data are available online for this figure.

shown that a complex comprising C9orf72, WDR41, and SMCR8 acts as a GEF for Rab8a and Rab39b in the autophagy pathway, thus providing functional evidence that C9orf72 is a *bona fide* DENN family protein with GEF activity (Sellier *et al*, 2016). On the other hand, we show here that C9orf72 binds to Rab1a (Fig 6) to regulate trafficking of the ULK1 complex during autophagy initiation (Figs 4 and 5). Moreover, we found that C9orf72 preferentially binds to GTP-loaded Rab1a, behavior expected of a Rab1a effector but not a Rab1a GEF (Fig 6D). It is well established that Rab GTPases act in so-called Rab GEF/GAP cascades in which the transition from an upstream Rab to a downstream Rab is achieved by recruiting the GAP and GEF for the upstream and downstream Rab GTPases as effectors, respectively (Hutagalung & Novick, 2011; Ao *et al*, 2014). Together, these data support the existence of a novel Rab GEF/GAP cascade, in which Rab1a as upstream Rab recruits C9orf72/SMCR8/WDR41 as a GEF of downstream Rab8a and Rab39b. As such, via interaction with the ULK1 complex, C9orf72 may link initiation of autophagy to downstream events. In agreement with such a model, SMCR8 and WDR41 were also found to associate with the ULK1 complex (Behrends *et al*, 2010; Sullivan *et al*, 2016).

C9orf72 is the second Rab GTPase associated protein involved in ALS. Alsin (ALS2), associated with juvenile ALS, is a Rab5 GEF (Topp *et al*, 2004). While Rab5 is best known for its vital role in the early endocytic pathway, similar to Rab1a, Rab5 also participates in autophagosome formation (Ao *et al*, 2014). Whereas Rab1a

regulates autophagosome formation via the Atg9/Atg2–Atg18 complex and as shown here regulates trafficking of the ULK1 initiation complex to the phagophore (Figs 4 and 5), Rab5 acts through the Vps34–Atg6/beclin1 class III PI3-kinase complex (Winslow *et al*, 2010; Lipatova & Segev, 2012; Dou *et al*, 2013; Wang *et al*, 2013). Thus, both C9orf72 and Alsin appear to be regulators of trafficking events in the autophagy pathway.

In the case of ALS2, disease is caused by loss of function of Alsin, suggesting defective autophagosome formation may be involved in ALS pathogenesis. Several groups have found that *C9orf72* mRNA levels are reduced in C9ALS/FTD (DeJesus-Hernandez *et al*, 2011; Cooper-Knock *et al*, 2012; Gijselinck *et al*, 2012, 2015; Belzil *et al*, 2013; Ciura *et al*, 2013; Donnelly *et al*, 2013; Mori *et al*, 2013b; Xi *et al*, 2013) and reduced C9orf72 protein levels have been detected in the frontal cortex of ALS and FTD cases (Waite *et al*, 2014; Xiao *et al*, 2015). Moreover, it has been reported that C9orf72 repeat size correlates with transcriptional downregulation of the promoter and onset age of disease (Gijselinck *et al*, 2015). Accordingly, C9orf72 loss of function by haploinsufficiency has been put forward as a possible cause of C9ALS/FTD, but this has been difficult to verify without knowing the cellular function of C9orf72. Here, we show that C9orf72 is required for the initiation of autophagy (Fig 1) and that C9ALS/FTD patient-derived iNeurons have significantly impaired basal levels of autophagy (Fig 9). Also in agreement with defective autophagy caused by C9orf72 loss of function, C9ALS/FTD

**Figure 7.  C9orf72 mediates interaction of Rab1a with the ULK1 complex.**

A   HEK293 cells treated with non-targeting (Ctrl) or C9orf72 siRNA were transfected with HA-ULK1, Myc-Rab1a, or HA-ULK1 + Myc-Rab1a as indicated. Transfections were laced with mVenus to enable identification of transfected cells for analysis (green). Transfected cells were probed with anti-HA and anti-Myc antibodies and processed for PLA. PLA proximity signals (red) per cell were determined from 3 independent experiments (mean ± SEM; one-way ANOVA with Fisher's LSD test, ****$P \leq 0.0001$; N (cells) = Ctrl/HA-ULK1: 125, Ctrl/Myc-Rab1a: 149, Ctrl/HA-ULK1 + Myc-Rab1a: 163, C9orf72/HA-ULK1: 136, C9orf72/Myc-Rab1a: 133, C9orf72/HA-ULK1 + Myc-Rab1a: 155). Scale bar = 20 µm.

B   HeLa cells treated with non-targeting (Ctrl) or C9orf72 siRNA were co-transfected with mCherry-FIP200 (red) and empty vector (EV) or Myc-Rab1a(Q70L) (green). Translocation of the ULK1 complex was quantified as the number of mCherry-FIP200-positive puncta per cell from 4 independent experiments (mean ± SEM; one-way ANOVA with Fisher's LSD test, *$P \leq 0.05$, ****$P \leq 0.0001$; N (cells) = Ctrl/EV: 101; Ctrl/Q70L: 106; C9orf72/Q70L: 42). C9orf72 knockdown was determined by RT–qPCR (Appendix Fig S2). Scale bar = 10 µm.

C   HeLa cells treated with non-targeting (Ctrl) or C9orf72 siRNA were co-transfected with EGFP-LC3 (green) and empty vector (EV) or Myc-Rab1a(Q70L) (red). Autophagosomes were quantified as the number of EGFP-LC3-positive puncta per cell (mean ± SEM; one-way ANOVA with Fisher's LSD test, **$P \leq 0.01$, ****$P \leq 0.0001$; N (cells) = Ctrl/EV: 44; Ctrl/Q70L: 32; C9orf72/Q70L: 47). C9orf72 knockdown was determined by RT–qPCR (Appendix Fig S2). Scale bar = 10 µm.

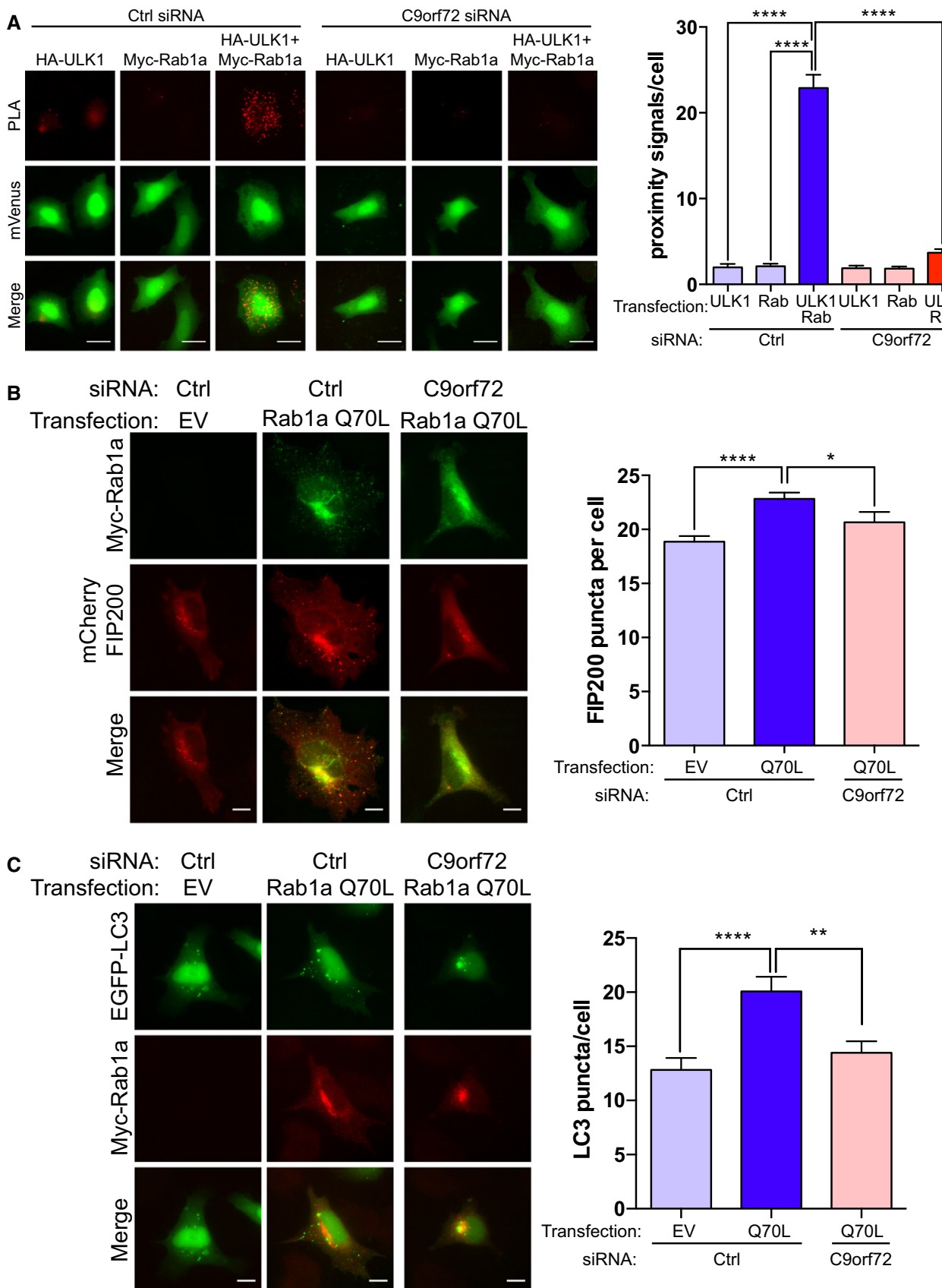

Figure 7.

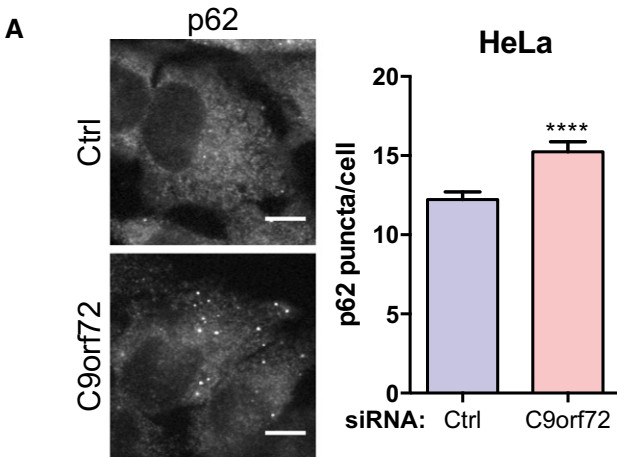

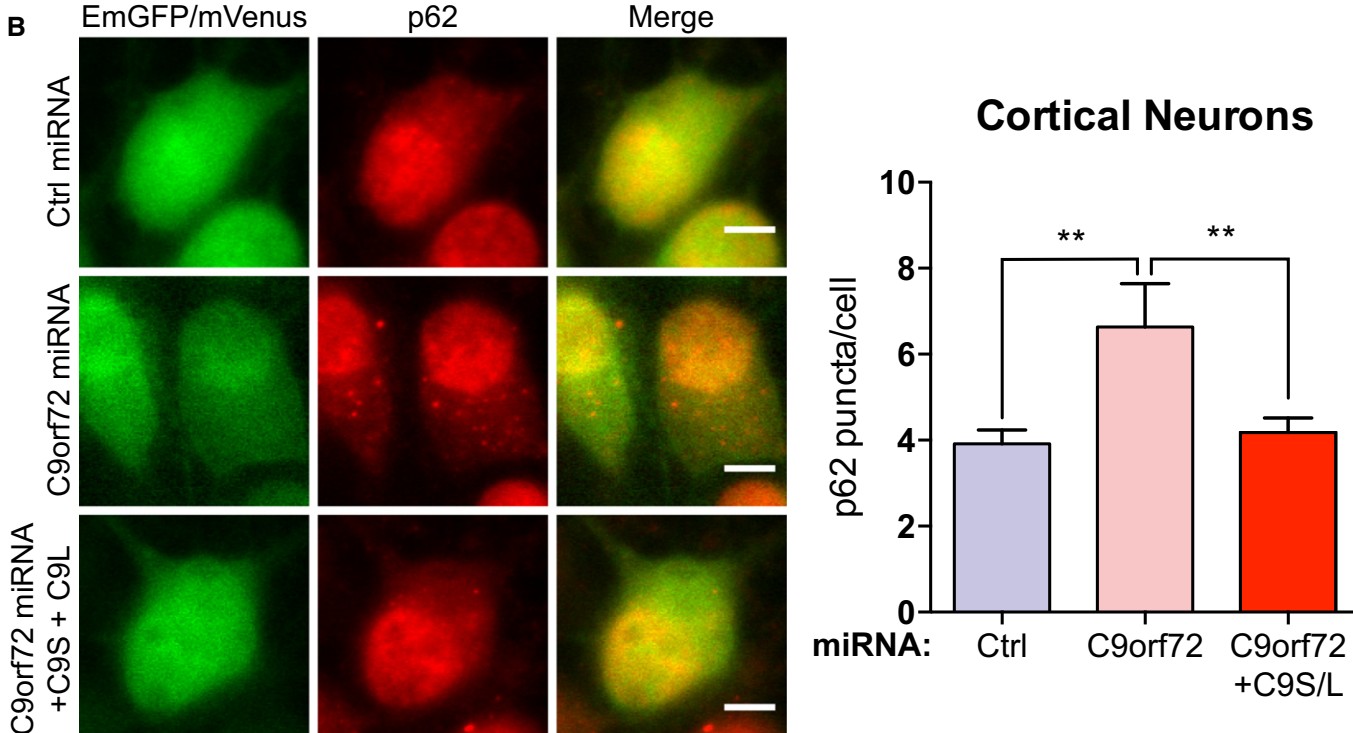

**Figure 8.  Loss of C9orf72 induces p62 accumulation.**

A   HeLa cells treated with non-targeting (Ctrl) or C9orf72 siRNA were immunostained for endogenous p62. Accumulation of p62 was quantified by counting p62-positive puncta per cell from 3 independent experiments (mean ± SEM; unpaired *t*-test, ****$P \leq 0.0001$; *N* (cells) = Ctrl: 74, C9orf72: 55). C9orf72 knockdown was confirmed by RT–qPCR (Appendix Fig S2). Scale bar = 10 μm.

B   Primary cortical neurons (DIV5) were transduced with 4TU/cell EmGFP non-targeting control miRNA (Ctrl) or C9orf72 miRNA (green); for rescue experiments, the cells were additionally transduced with 4TU/cell mVenus-tagged C9orf72s and C9orf72L (verified by immunoblot, Appendix Fig S2). Neurons were immunostained for endogenous p62 3 days post-transduction. Accumulation of p62 was quantified by counting p62-positive puncta per soma from 2 independent experiments (mean ± SEM; one-way ANOVA with Fisher's LSD test, **$P \leq 0.01$; *N* (cells) = Ctrl miRNA: 80; C9orf72 miRNA: 80; C9orf72 miRNA+C9orf72L+C9orf72S: 75). Scale bar = 10 μm.

patients show specific ubiquitin and p62-positive inclusions in the brain (Al-Sarraj *et al*, 2011; Cooper-Knock *et al*, 2012; Mackenzie *et al*, 2014) and our data show that loss of C9orf72 in cells and primary neurons mimics C9ALS/FTD p62 pathology (Fig 8). Thus, our findings provide a molecular explanation for C9ALS/FTD p62 pathology and suggest that autophagy deficits due to C9orf72

haploinsufficiency may contribute to C9ALS/FTD. Clearly, more work is required to assess the contribution of defective autophagy to neuronal loss in C9ALS/FTD, but it is well established that loss of neuronal autophagy can cause neurodegeneration. Neuron-specific knockout of the essential autophagy genes *ATG7*, *ATG5*, and *RB1CC1* (FIP200) in mice causes neurodegeneration, progressive

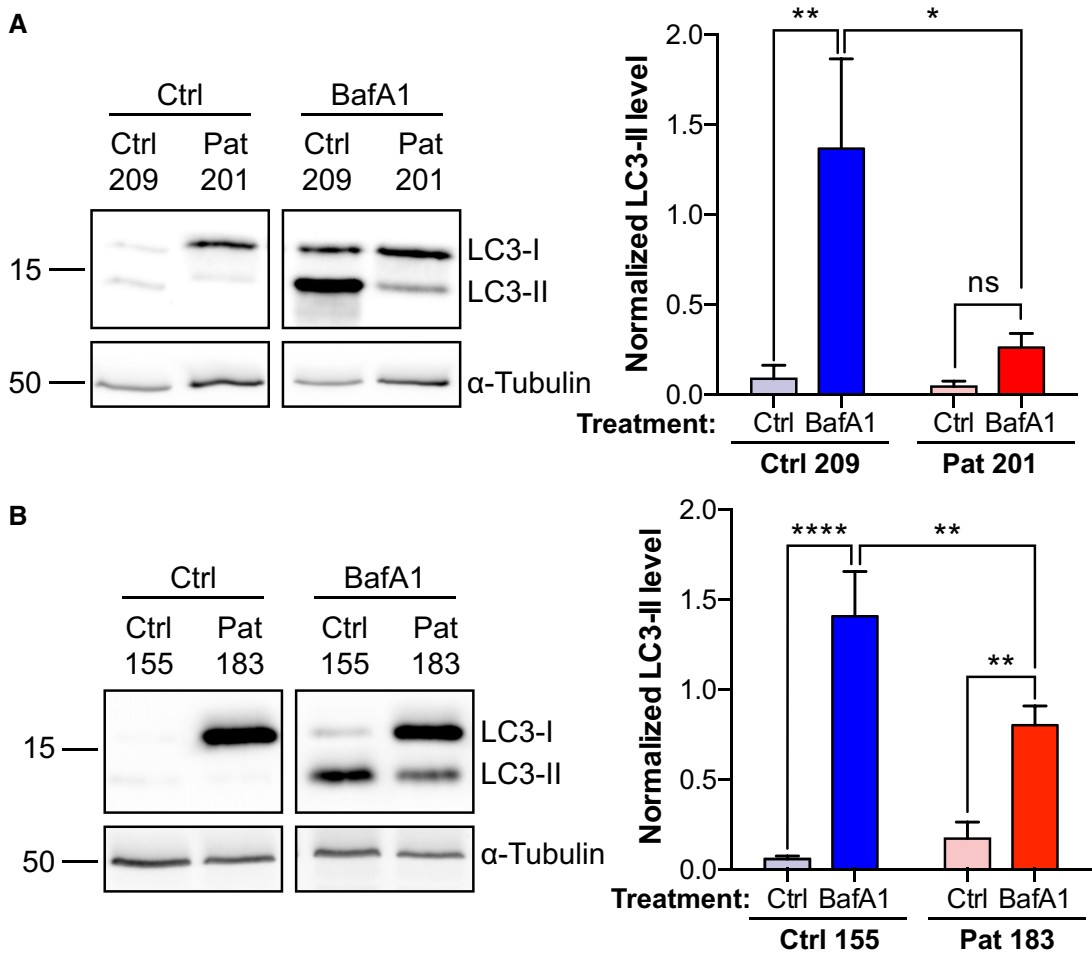

**Figure 9. C9ALS/FTD iNeurons exhibit autophagy deficits.**

A, B   Two C9ALS/FTD iNeuron cultures (201 in A and 183 in B) and their matching controls (209 in A and 155 in B) were treated with vehicle (Ctrl) or bafilomycin A1 (BafA1, 100 nM; 6 h) and processed for immunoblot detection of LC3. LC3-II levels were normalized to α-tubulin (mean ± SEM; one-way ANOVA with Fisher's LSD test; ns, not significant; *$P \leq 0.05$; **$P \leq 0.01$; ****$P \leq 0.0001$; Ctrl 209/Pat 201, $n = 4$; Ctrl 155/Pat 183, $n = 6$).

Source data are available online for this figure.

deficits in motor function, including abnormal limb-clasping reflexes (as also observed in ALS-SOD1 mice) and a reduction in coordinated movement, that are accompanied by the accumulation of cytoplasmic inclusion bodies in neurons (Hara *et al*, 2006; Komatsu *et al*, 2006; Liang *et al*, 2010). In these neuronal autophagy-deficient mice, the cerebellum is particularly affected suggesting that the cerebellum is singularly reliant on autophagy (Hara *et al*, 2006; Komatsu *et al*, 2006; Liang *et al*, 2010). Similarly, in C9ALS/FTD patients ubiquitin/p62-positive/TDP43-negative pathology appears to be largely restricted to the cerebellum and hippocampus (Al-Sarraj *et al*, 2011; Cooper-Knock *et al*, 2012; Mackenzie *et al*, 2014). Knockout of *C9orf72* in mice did not cause motor neuron degeneration or motor deficits (Koppers *et al*, 2015; Atanasio *et al*, 2016; O'Rourke *et al*, 2016; Sudria-Lopez *et al*, 2016; Sullivan *et al*, 2016), indicating that in contrast to *ATG7, ATG5*, and *RB1CC1* (FIP200), *C9orf72* is not essential for all autophagy in mice. Alternatively, there may be redundancy of the *C9orf72* gene in mammalian species. Full knockout of *C9orf72* did cause altered immune

responses in macrophages and microglia, suggesting C9orf72 regulates immune homeostasis (Atanasio *et al*, 2016; O'Rourke *et al*, 2016; Sudria-Lopez *et al*, 2016; Sullivan *et al*, 2016). The role of autophagy in the immune system is well established (Shibutani *et al*, 2015); thus, possibly C9orf72 plays a specific role in immune-related autophagy initiation. Consistent with defective autophagy, these *C9orf72* knockout mice showed lysosomal accumulation and increased amounts of p62 and both LC3-I and II in homozygote spleens (O'Rourke *et al*, 2016; Sullivan *et al*, 2016). However, since both LC3-I and II were increased, this is not likely to be due to a block in late autophagy as suggested by O'Rourke *et al* (2016) but rather a compensatory increase in response to decreased autophagy initiation. Indeed, we observed a similar increase of LC3-I in patient-derived iNeurons (Fig 9) and in *C9orf72* knockout zebrafish and mice (data not shown).

Impaired autophagy has also been implicated in non-C9orf72 ALS/FTD and other neurodegenerative diseases (Chen *et al*, 2012; Harris & Rubinsztein, 2012). Abnormal autophagy has been found

in SOD1 and TDP43 models of ALS, and as mentioned above, Alsin may regulate autophagy via Rab5. Furthermore, several ALS/FTD genes, such as charged multivesicular body protein-2B (CHMP2B), valosin-containing protein (VCP), ubiquilin 2, p62 (sequestosome 1), optineurin, and TANK-binding kinase (TBK1), are directly linked to autophagy. Thus, neurodegeneration associated with defective neuronal autophagy appears to be intimately linked with ALS/FTD, reinforcing autophagy as a possible therapeutic target for ALS/FTD.

# Materials and Methods

### Plasmids

C9orf72L and C9orf72S cDNA was generated by PCR using Phusion High Fidelity enzyme (NEB) from a HEK293 cDNA library prepared in house, using primers containing restriction sites. The Myc- and FLAG-tagged constructs were cloned by standard subcloning techniques into pRK5-Myc and p3xFLAG-CMV, using SalI/NotI and HindIII/BamHI restriction sites, respectively. GST-tagged C9orf72S and C9orf72L constructs were generated by subcloning into pGEX6p1 (GE Healthcare Life Sciences) using XhoI/NotI sites. EGFP-tagged C9orf72S and C9orf72L were generated by EcoRI/BamHI subcloning into pEGFPc2. mVenus-tagged C9orf72S and C9orf72L were generated by XhoI/NotI subcloning into pCI-Neo-mVenus-N, a modified pCI-neo vector containing mVenus cDNA, generated by PCR using Phusion High Fidelity enzyme from prSETB-mVenus (a gift from Atsushi Miyawaki, RIKEN, Japan). mVenus-C9orf72 was subcloned into pLvos, a modified pL-SIN-PGK-cPPT-GDNF-WHV lentiviral vector (Déglon *et al*, 2000; Suzuki *et al*, 2007) in which the original NheI and NotI sites were destroyed and the GDNF cDNA was replaced by a pCI-neo-derived multiple cloning site (5′-gat cct aat acg act cac tat agg gcg gct agc ctc gag aat tca cgc gtc gcc ggc gca tat gct gca gcc cgg gcg gcc gct tcc ctt tag tga ggg tta atg-3′) using the NheI/NotI sites to yield pLvos-mVenus-C9orf72. pL-SIN-PGK-cPPT-EGFP-WHV was a kind gift from M. Azzouz, SITraN, Sheffield UK (Nanou *et al*, 2013). p3xFLAG-CMV10-hFIP200 was a gift from Noboru Mizushima, Tokyo Medical and Dental University, Japan, via Addgene (plasmid #24300) (Hara *et al*, 2008). FIP200-6xHis fragments were generated by PCR using Phusion High Fidelity enzyme (primers: FIP200-FW: ctcgaggccacc-*atgaagttatatgtatttctggtta* FIP200-638-R gcggccgctcaatggtggtggtgatgatg-*ttcactcagtagatctgtaatg* FIP200-639-FW ctcgaggccaccatg-*caaaaggcatctgtgagtcag* FIP200-1373-6His-R gcggccgctcaatggtggtggtgatgatg-*ttcagaaagtgactctatcaaat* FIP200-1374-FW ctcgaggccaccatg-*gatcgagctcgtttgcttgag* FIP200-6His-R gcggccgct-caatggtggtggtgatgatg-*ttatactttcttattccatgatacg*) and transferred to pCi-Neo (Promega) using XhoI/NotI sites. mCherry-FIP200 was a gift from Xuejun Jiang, Memorial Sloan-Kettering Cancer Center, NYC, USA (Ganley *et al*, 2009). pRK5-HA-ULK1 and pRK5-Myc-hATG13 were a gift from Do-Hyung Kim, University of Minnesota, USA, via Addgene (plasmid #31963 and #31965) (Jung *et al*, 2009). pCMV-intron Myc-Rab1aWT and Myc-Rab1aS25N were from T. Hébert, McGill University, Canada, via Addgene (plasmid #46776 and #46777) (Dupre *et al*, 2006). Rab1a(Q70L) was generated from Rab1aWT by site-directed mutagenesis using a QuikChange Lightning Site-Directed Mutagenesis Kit (Agilent) according to the manufacturer's instructions using 5′-tgttcgaaatctttccaggcctgctgtgtccc-3′ and 5′-gggacacagcaggcctggaaagatttcgaaca-3′ primers. mCherry-EGFP-LC3b

was a gift from Terje Johansen, University of Tromsø, Norway (Pankiv *et al*, 2007). EGFP-LC3b was a gift from Chris Miller, KCL, UK. All constructs were confirmed by sequencing.

### Cell culture and transfection

HEK293, HeLa, and SH-SY5Y cells (ATCC) were cultured in Dulbecco's modified Eagle's medium (DMEM, Sigma) supplemented with 10% FBS (Biosera) and 1 mM sodium pyruvate (Sigma) in a 5% $CO_2$ atmosphere at 37°C. Cells were transfected with plasmid DNA using Lipofectamine 2000 reagent (Invitrogen) according to the manufacturer's instructions or polyethylenimine (PEI) (stock 1 mM; 3 μl/μg plasmid). HeLa, HEK293, and SH-SY5Y cells were siRNA transfected using Lipofectamine RNAiMax or Lipofectamine 2000 (Invitrogen) according to the manufacturer's instructions. Cells were used for experiments 4 days after siRNA transfection.

Cortical neurons were isolated from E18 Sprague Dawley rat embryos (Charles River) and cultured on glass coverslips coated with poly-L-lysine in 12- or 24-well plates in neurobasal medium supplemented with B27 supplement (Invitrogen), 100 IU/ml penicillin, 100 mg/ml streptomycin, and 2 mM L-glutamine. Cortical neurons (DIV5–6) were transfected using Lipofectamine LTX with PLUS reagent according to the manufacturer's instructions (Thermofisher). Briefly, neurons were placed in fresh culture medium with a DNA:PLUS:LTX ratio of 1:0.5:0.5 (0.5–1 μg DNA/100,000 cells/cm$^2$). After 6 h, the transfection mix was replaced with conditioned medium. For lentiviral transduction, neurons were exposed to 4TU/cell in fresh culture medium overnight.

### iNPC production and neuronal differentiation

Induced neural progenitor cells (iNPCs) were derived from human skin fibroblasts as previously described (Meyer *et al*, 2014). Human skin fibroblast samples were obtained from PJS from the Sheffield tissue bank. Informed consent was obtained from all subjects before sample collection (Study number STH16573, Research Ethics Committee reference 12/YH/0330). Briefly, 10,000 fibroblasts were transduced with lentiviral vectors for OCT3, Sox2, KLF4, and C-MYC for 12 h. Forty-eight hours after transduction, the cells were washed with PBS and fibroblast medium was replaced with NPC medium (DMEM/F-12 with glutamax supplemented with 1% N2, 1% B27, 20 ng/ml FGF-b, 20 ng/ml EGF, and 5 μg/ml heparin. When the cells started changing shape and form neurospheres, they were expanded as neural rosettes. When the iNPC culture was confluent (~3 weeks), EGF and heparin were withdrawn and the FGF-b concentration increased to 40 ng/ml. The iNPCs can be maintained for ~30 passages. iNPCs are not expanded by clone and therefore do not display clonal variability.

For differentiation, 30,000 iNPCs were plated in a 6-well plate coated with fibronectin (Millipore) and expanded to 70-80% confluence after which iNPC medium was replaced with neuron differentiation medium (DMEM/F-12 with glutamax supplemented with 1% N2 and 2% B27). On day one of differentiation, the cells were treated with 2.5 μM DAPT (Tocris) to determine differentiation toward a neuronal lineage. On day three, the neuron differentiation medium was supplemented with 1 μM retinoic acid (Sigma), 0.5 μM smoothened agonist (SAG) (Millipore) and 2.5 μM forskolin (Sigma) for 7 days. This protocol leads to typical yields of 70% β-III tubulin

(Tuj1)-positive cells. iNeuron experiments were repeated at least 3 times for each sample starting from iNPCs at different passages, typically between passages 10 and 20 to ensure that the results observed are not determined by batch effects or passage number.

### siRNA and miRNA

Non-targeting control siRNA and targeting siRNA were purchased from Sigma. The siRNA sequences were as follows: C9orf72#2: gugcuauagauguaaaguu, C9orf72#D: gaucaggguucagaguauua, FIP200#1 caaguuagagguugaacuu, FIP200#2 gaucuuaugugaucgucca, Rab1a#1 gaacaaucaccuccaguua, Rab1a#2 cagaucaggaguccuucaa.

miRNA targeting rat C9orf72 (miR$^{C9orf72}$) was designed using Invitrogen's RNAi Designer (http://rnaidesigner.thermofisher.com/rnaiexpress/) and cloned into the pcDNA6.2™-GW/EmGFP miR expression vector according to the manufacturer's instructions (BLOCK-iT™ Pol II miR RNAi expression vector kit, Invitrogen; top strand: 5′-tgc tgt atc cca gta agc aaa ggt agg ttt tgg cca ctg act gac cta cct ttt tac tgg gat a-3′). pcDNA6.2™-GW/ECFP miR was generated by replacing EmGFP with ECFP in pcDNA6.2™-GW/EmGFP. EmGFP miR$^{C9orf72}$ was amplified from pcDNA6.2™-GW using 5′-gct agc aca agt ttg tac aaa aaa gca-3′ and 5′-gcg gcc gcc aac cac ttt gta caa gaa agc t-3′ primers and subcloned into pLvos using XhoI/NotI sites. Non-targeting miRNA is part of the BLOCK-iT™ Pol II miR RNAi expression vector kit.

### Lentiviral production

$3 \times 10^6$ HEK293T cells (ATCC) were seeded per 10-cm dish and transfected with 12.5 μg of DNA the following day (4.95 μg of pL expression constructs, 4.95 μg pCMV-dR8.92 (a gift from Bob Weinberg via Addgene (plasmid #8455)) (Stewart *et al*, 2003), 1.45 μg pMD2.G, and 1.15 μg pRSV-Rev (both from Didier Trono via Addgene (plasmid #12253 and #12259) (Dull *et al*, 1998)) using polyethyleneimine (PEI, 1 mg/ml; PolySciences) at a ratio of 1:3 (DNA:PEI). Cells were incubated for 72 h, and virus particles were harvested by ultracentrifugation of the cell culture supernatants (90 min at 65,000 *g*, rotor SW-28, Optima™ L-100 K Ultracentrifuge, Beckman Coulter). Viral pellets were resuspended in 1% BSA/PBS, and aliquots were stored at −80°C. In order to titer the viruses, HeLa cells were seeded into 12-well plates at a density of 75,000 cells per well and transduced with 500 μl of a $10^{-2}$, $10^{-3}$, and $10^{-4}$ dilution of the virus the next day. After 72 h, cells were trypsinized and fixed (4% formaldehyde/PBS) and EGFP or mVenus fluorescence was analyzed by fluorescence-activated cell sorting (FACS; BD™ LSR II Flow Cytometer). The following formula was used to calculate the number of TU/ml: number of cells before transduction/percentage of positive gated cells × dilution factor × 2.

### SDS–PAGE and immunoblotting

Cells were harvested in trypsin/EDTA (Lonza) and pelleted at 400 *g* for 4 min. Pellets were washed once with phosphate-buffered saline (PBS). Cells were lysed on ice for 30 min in ice-cold RIPA buffer (50 mM Tris–HCl pH 6.8, 150 mM NaCl, 1 mM EDTA, 1 mM EGTA, 0.1% (w/v) SDS, 0.5% (w/v) deoxycholic acid, 1% (w/v) Triton X-100, and protease inhibitor cocktail (Thermo Scientific)). Lysates were clarified at 15,000 *g* for 30 min at 4°C. Protein concentration

was measured by Bradford protein assay (Bio-Rad). Proteins were separated by SDS–PAGE and transferred to nitrocellulose membranes (Whatmann) by electroblotting (Bio-Rad). After transfer, membranes were blocked for 1 h at room temperature in Tris-buffered saline (TBS) with 5% fat-free milk (Marvel) and 0.1% Tween-20. Membranes were incubated with primary antibodies in blocking buffer for 1 h at room temperature or overnight at 4°C. Membranes were washed 3 times for 10 min in TBS with 0.1% Tween-20 before incubation with secondary antibodies in block buffer for 1 h at room temperature. After washing, membranes were prepared for chemiluminescent signal detection with SuperSignal West Pico Chemiluminescent substrate (Thermo Scientific) according to the manufacturer's instructions. Signals were detected with a SynGene Gbox or on ECL film (GE Healthcare). Signal intensities were quantified using ImageJ (Abramoff *et al*, 2004).

### Immunoprecipitation

Cells were harvested in trypsin/EDTA and pelleted at 400 *g* for 4 min. Pellets were washed once with phosphate-buffered saline (PBS). Cells were lysed at 4°C for 1 h in ice-cold BRB80 lysis buffer (80 mM K-PIPES pH 6.8, 1 mM MgCl$_2$, 1 mM EDTA, 1% (w/v) NP-40, and protease inhibitor cocktail). Lysates were clarified at 15,000 *g* for 30 min at 4°C. 2 mg of protein was incubated with 2 μg of primary antibody for 16 h at 4°C. Antibody was then captured by incubation with 25 μl of 50% protein G-Sepharose (Sigma) for 2 h at 4°C. After centrifuging at 3,000 *g*, immune pellets were washed 5 times in ice-cold lysis buffer. Protein was eluted from protein G beads in 2× Laemmli buffer. Samples were analyzed by SDS–PAGE and immunoblot.

### *In vitro* binding assays

[$^{35}$S]-methionine-labeled FIP200 fragments, HA-ULK1, Myc-ATG13, and Myc-Rab1a were produced using the T7/SP6 TnT Quick Coupled Transcription/Translation kit (Promega). GST, GST-C9orf72S, and GST-C9orf72L were expressed from the pGEX6p1 vector in Rossetta PLysS cells. 0.1 g GST cell pellet and 0.25 g GST-C9orf72 cell pellets were lysed by sonication in 1 ml of buffer RB100 (25 mM Hepes (pH 7.5), 100 mM KOAc, 10 mM MgCl$_2$, 1 mM DTT, 0.05% Triton X-100, 10% (vol/vol) glycerol), and GST-tagged proteins bound to 30 μl of glutathione-agarose (GSH) bead slurry. Binding reactions were carried out in 400 μl buffer RB100 containing the 30 μl GSH beads and bound proteins along with 8 μl of labeled protein. In case of Myc-Rab1a, 1 mM GDP or 1 mM GMP-PNP (guanosine 5′-[β,γ-imido]triphosphate trisodium salt hydrate) was added to the binding assays as indicated. Proteins were eluted from GSH beads using glutathione elution buffer (50 mM Tris–HCl (pH 7.5), 100 mM NaCl, 40 mM reduced glutathione) and analyzed by SDS–PAGE. 1 μl reticulocyte lysate was loaded as input. Cells were stained with Coomassie to show GST-tagged proteins and $^{35}$S radiolabel was detected using a Typhoon Phosphorimager (GE Healthcare).

### Antibodies

Primary antibodies used were as follows: rabbit anti-human Tuj1 (Covance, IF: 1:500), mouse anti-FLAG (M2, Sigma, WB: 1:2,000,

IF: 1:1,000), mouse anti-Myc (9B11, Cell Signaling, WB: 1:2,000, IF: 1:2,000), mouse anti-tubulin (DM1A, Sigma, WB: 1:10,000), rabbit anti-Myc (ab9106, Abcam, WB: 1:2,000, IF: 1:1,000), rabbit anti-GAPDH (14C10, Cell Signaling, WB: 1:2,000), rabbit anti-LC3 (2220, Novus Biologicals, WB: 1:1,000), mouse anti-p62 (610833, BD Biosciences, IF: 1:1,000), rabbit anti-p62 (18420-1-AP, ProteinTech, IF: 1:200), rabbit anti-FIP200 (SAB4200135, Sigma, WB: 1:500), rabbit anti-ULK1 (#8054, Cell Signaling, WB: 1:1,000), rabbit anti-phospho-ULK S757 (#6888, Cell Signaling, WB: 1:1,000), mouse anti-EGFP (JL8, Clontech, WB: 1:5,000), rabbit anti-ATG13 (#6940, Cell Signaling, WB: 1:1,000), rabbit anti-Rab1a (Ab97956 Abcam, WB: 1:1,000), and rabbit anti-HA (Sigma, WB: 1:2,000, IF: 1:1,000). Secondary antibodies used for immunoblotting were horseradish peroxidase-coupled goat anti-rabbit, goat anti-rat, and rabbit anti-mouse IgG (Dako; 1:5,000). Secondary antibodies used for immunofluorescence were Alexa fluorophore (488, 568, or 633)-coupled goat/donkey anti-mouse IgG, Alexa fluorophore (488, 568, or 633)-coupled goat/donkey anti-rabbit IgG (Invitrogen; 1:500).

### RNA extraction and RT–qPCR

RNA was extracted from HEK293, HeLa, SH-SY5Y, rat cortical neurons, or iNeuron cells using TRIzol reagent (Invitrogen), according to the manufacturer's instructions. Extracted RNA was dissolved in 20 μl of nuclease-free water. RNA from whole-cell extraction was reverse-transcribed into cDNA using Superscript III reverse transcriptase (Invitrogen). 2 μg RNA was transcribed in a final volume of 10 μl with 2 μl of 25 mM dNTP mix (NEB), 4 μl of 5× reverse transcriptase buffer (Invitrogen), 2 μl of 0.1 M DTT, 1 μl of Superscript III, and 1 μl of Oligo(dT) (Thermo Scientific). RT–qPCR was performed using the Stratagene Mx3000P and MxPro v4.10 software. Samples were amplified in triplicate in 20 μl volumes using SYBR-Green master mix and 250 nM of each optimized forward and reverse primer. Cycling conditions for RT–qPCR were as follows: 95°C for 10 min to denature followed by 35 cycles of 95°C for 30 s, 60°C for 1 min. Levels of mRNA were quantified relative to GAPDH mRNA levels according to the $\Delta\Delta Ct$ method.

Primer sequences were as follows: human C9orf72 (all isoforms), FW gttgatagattaacacatataatccgg, REV cagtaagcattggaataatactctga; human Rab1a, FW tgtccagcatgaatcccgaa, REV ggcaagactttccaacccct; human GAPDH, FW gggtggggctcatttgcaggg, REV tgggggcatcagca gagggg; rat GAPDH FW: tgaagggtggggccaaaagg, REV ggtcatgagcccttc catga; and rat C9orf72 FW: gtgttgacaggctaacgcac, REV: agggatgacc tccccagtaa.

### Induction of autophagy

Autophagy was induced by incubation with 500 nM rapamycin for 6 h or 250 nM Torin1 for 3 h. Autophagy was blocked by addition of 100 nM bafilomycin A1 for 5 (neurons) or 6 h. Where cells were treated with Torin1 and bafilomycin A1, Torin1 was added 2 (neurons) or 3 h after addition of bafilomycin A1.

### Yeast 2 hybrid

C9orf72S was cloned as "bait" fused to the DNA binding domain of GAL4 in pGBKT7 using EcoRI/BamHI restriction sites and was used to screen a human brain random cDNA library. Screens were

performed by Protein Interaction Screening, Genomics and Proteomics Core Facilities (W150), German Cancer Research Center (DKFZ), Heidelberg, Germany.

### Immunofluorescence

Immunostaining was performed as described previously (De Vos *et al*, 2005). Briefly, cells grown on glass coverslips were fixed with 3.7% formaldehyde in phosphate-buffered saline (PBS) for 20 min at room temperature. After washing with PBS, residual formaldehyde was quenched by incubation with 50 mM $NH_4Cl$ in PBS for 10 min at room temperature, followed by a second round of washing with PBS. Subsequently, the cells were permeabilized by incubation with 0.2% Triton X-100 in PBS for 3 min. Triton X-100 was removed by washing with PBS. After fixing, the cells were incubated with PBS containing 0.2% fish gelatin (PBS/F) for 30 min at room temperature (cell lines) or PBS containing 2% fish gelatin for 1-2 h (neurons) and then with the primary antibody in PBS/F for 1 h at RT or ON at 4°C. After washing with PBS/F, the cells were incubated with secondary antibody in PBS/F for 45 min at room temperature and stained with Hoechst 33342. After a final wash, the samples were mounted in fluorescence mounting medium (Dako).

Images were recorded using appropriate filtersets (Omega Optical and Chroma Technology) using MicroManager 1.4 software (Edelstein *et al*, 2014) on a Zeiss Axioplan2 microscope fitted with a Hamamatsu C4880-80 or Retiga R3 (QImaging) CCD camera, PE-300 LED illumination (CoolLED), and a 63×, 1.4NA Plan Apochromat objective (Zeiss) or on a Zeiss Axiovert 200 microscope equipped with a Hamamatsu C9100-12 EMCCD, PE-4000 LED illumination (CoolLED), and 63×, 1.4NA Plan Apochromat and 100×, 1.3NA Plan Apochromat objectives (Zeiss) and using MetaMorph software (Universal Imaging) on an Olympus IX83 equipped with a Zyla4.2 sCMOS camera (Andor), SpectraX light engine (Lumencor) and OptoLED (Cairn Research) illumination, and 60×, 1.35NA Universal Plan Super Apochromat and 40×, 1.35NA Universal Apochromat objectives (Olympus). Illumination intensities, exposure times, and camera settings were kept constant during experiments.

### Proximity ligation assay

*In situ* proximity ligation assays were performed with the Duolink *In Situ* Kit following the manufacturer's protocol (Olink Bioscience). Twenty-four hours post-transfection, cells were fixed, quenched, and permeabilized. Samples were blocked according to the manufacturer's instructions. Samples were then probed with primary antibodies for 1 h at room temperature. After washing, cells were incubated with mouse-minus and rabbit-plus PLA probes according to the manufacturer's instructions for 1 h at 37°C. After further washes, the PLA probes were ligated for 30 min at 37°C before the amplification step was performed for 100 min at 37°C using the Duolink Orange detection reagents. After final washes, the samples were mounted in fluorescence mounting medium (Dako).

### Image analysis

All image analysis was performed using ImageJ (Abramoff *et al*, 2004). mCherry-FIP200, EGFP-LC3, mCherry-EGFP-LC3, p62 puncta, and PLA proximity signals were counted in single cells using the

Particle Analysis facility of ImageJ. Where possible, the cells for analysis were selected based on fluorescence in the other channel, else the samples were blinded to the operator. Images were filtered using a Hat filter ($7 \times 7$ kernel) (De Vos & Sheetz, 2007) to extract puncta and proximity signals and thresholded such that the visible puncta within the cell were highlighted, but no background was included. The result of thresholding was further checked against the original image to ensure no background signal was identified as puncta or signals were lost. In case of mCherry-EGFP-LC3, puncta in the red and green channels were counted separately. Red-only puncta were determined by subtracting the green puncta from the red puncta.

### Statistical analysis

Calculations and statistical analysis were performed using Excel (Microsoft Corporation, Redmond, WA) and Prism 6 software (GraphPad Software Inc., San Diego, CA).

**Expanded View** for this article is available online.

### Acknowledgements

We thank Noburo Mizushima (Tokyo Medical and Dental University, Japan), Xuejun Jiang (Memorial Sloan-Kettering Cancer Center, NYC, USA), Do-Hyung Kim (University of Minnesota), T Hébert (McGill University, Canada), Terje Johansen (University of Tromsø, Norway), and Chris Miller (KCL, UK) for the sharing of reagents. This work was funded by grants from the Thierry Latran Foundation (Project RoCIP to KJDV, AJG, PJS), Medical Research Council (MRC) (MR/K005146/1 to KJDV and MR/M013251/1 to KJDV, AJW), Alzheimer's Society (260 (AS-PG-15-023) to KJDV, CPW, AJG), the University of Sheffield Moody Family Endowment (to KJDV, CPW), and a EU Framework 7 Award (Euromotor No 259867 to PJS). PJS is supported as an NIHR Senior Investigator. EFS is supported by a Motor Neurone Disease Association Prize Studentship (DeVos/Oct13/870-892 to KJDV, AJG). LF is supported by a Marie Curie International Fellowship (EU framework 7 grant 303101).

### Author contributions

CPW, EFS, CSB, LF, MM, GMH, AH, MJW, AM, AJG, and KJDV performed experiments. BKK, KM, and PJS provided reagents. AJG and KJDV designed experiments. CPW, EFS, CSB, LF, AM, and KJDV wrote the manuscript. GMH, PJS, AJW, BKK, KM, and AJG edited the manuscript. KJDV supervised the research.

### Conflict of interest

The authors declare that they have no conflict of interest.

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
