## [Review Process File · The EMBO Journal]

Manuscript EMBO-2016-94401

The C9orf72 protein interacts with Rab1a and the ULK1 complex to regulate initiation of autophagy

Christopher P Webster, Emma F Smith, Claudia S Bauer, Annekathrin Moller, Guillaume M Hautbergue, Laura Ferraiuolo, Monika A Myszczyńska, Adrian Higginbottom, Matthew J Walsh, Alexander J Whitworth, Brian K Kaspar, Kathrin Meyer, Pamela J Shaw, Andrew J Grierson, Kurt J De Vos

Corresponding author: Kurt De Vos, The University of Sheffield

Review timeline:	Submission date:	22 March 2016
	Editorial Decision:	22 April 2016
	Revision received:	23 May 2016
	Editorial Decision:	01 June 2016
	Revision received:	02 June 2016
	Accepted:	02 June 2016

Editor: Karin Dumstrei

Transaction Report:

1st Editorial Decision

22 April 2016

Thank you for submitting your manuscript to the EMBO Journal. Your study has now been seen by two referees and their comments are provided below.

As you can see from the comments your manuscript received a mixed response. While both referees appreciate the insights made, they also raise significant concerns with the analysis. They find that key findings have to be extended to primary neuronal cells and that more support is needed for the proposed mechanism. As also indicated by referee #2 (and as you might also have seen) we published yesterday a study reporting a role for c9orf72 in autophagy, but via a different mechanism (Sellier et al.). Given this related study, timing is an important issue in this case.

What I can offer is that should you be able to address the major points raised by referee #1 within 4 weeks that I can offer to consider a revised version. The revised version would have to extend the key findings to primary neuronal cells (point 1), to add more support for the mechanism (point 2 and 4) and to do the siRNA rescue experiment (point 3).

I know that this is a tall order to do this within 4 weeks, but as mentioned above timing is very important given the publication of the related paper. The revised version would have to be re-reviewed by the referees and I need endorsement from them to consider publication here.

I recognise that it might not be possible for you to address the raised concerns within 4 weeks and if

that is the case you should seek rapid publication elsewhere at this stage.

It would be good if you could let me know what you decide to do. If you have further questions feel free to contact me.

REFEREE REPORTS

Referee #1:

Webster, Smith and collaborators

general summary

This manuscript by Webster, Smith and colleagues investigates the very interesting and timely question of the function of the C9ORF72 protein. Indeed, the consensus is that repeat expansions trigger decreased expression of C9ORF72 gene, yet the consequences of this haploinsufficiency remain elusive (and controversial). Here the authors used mostly cell lines to perform loss and gain of function experiments on the C9ORF72 protein and address a potential function in autophagy. This initial hypothesis is seducing due to the undisputed existence of strong p62 pathology in C9+ cases. The authors show that knocking down C9ORF72 impairs autophagy initiation, and provide data suggesting that this occurs through direct effects on the translocation of FIP200/ULK1 complex during autophagy initiation via Rab1a. Importantly, the authors show that knocking down C9ORF72 leads to p62 accumulation, reminiscent of what occurs in patients. Last, preliminary results in induced neurons seem to confirm a role for C9ORF72 in autophagy. In general, the study is timely and appear well designed in its conception. There are however major drawbacks that decrease the enthusiasm.

Major concerns

- 1) Most of the studies were done in cell lines that are not even neuronal (HEK293 and Hela cells). Autophagy is notoriously important in neurons, and likely to be regulated by different pathways/effectors than in peripheral cells. Thus, the key data should be reproduced in (ideally primary) neuronal cells. It is worth mentioning that some data are reproduced in induced neurons, but this is not the case of Rab1a or FIP200 involvement.
- 2) the interaction between C9ORF72 and the ULK1 complex remains poorly documented. The authors perform 2 sets of experiments, each with their weaknesses. First co-Ips are performed only on transfected, overexpressing cells. While C9ORF72 are notoriously poor, the authors could at least show that endogenous FIP200, ULK1 and Rab1a interact with transfected C9ORF72 isoforms. In the same line, it is not clear while some non specific bands are present only in GST-C9ORF72 wells in Figure 3B, and not in the GST control. It seems also that C9ORF72L is present in the recombinant GST-C9ORF72S preparation ? To document a true interaction, the authors should be more quantitative, and perform competition assays, vary salinity conditions, or, even better measure interaction constants with BiaCore or similar setup. Alternatively, and even if very indirect, a PLA assay between C9ORF72 and Ulk1 members would bring additional evidence. As such, I am not convinced that C9ORF72 and members of the ULK1 complex are true interactors in vivo.
- 3) the specificity of the knockdown of C9ORF72 effects could be shown to be rescued by a siRNA resistant form of C9ORF72. This is especially important in induced neurons as these neurons also have potential RNA toxicity and DPR toxicity, that could in principle also interfere with autophagy.
- 4) the authors show that overexpression of constitutively active Rab1a reverts the effects on FIP200 punctae. This suggests that Rab1a activation is sufficient to revert this defect due to C9ORF72 knock down. However, and most importantly, the authors do not document the effects on LC3 and autophagosome formation in the same experimental conditions (although the magnitude of the expected effects is quite larger - compare Figure 1b with Figure 4b). It would also be important to perform the converse experiment, that is to show that Rab1a dominant negative expression inhibits increased autophagic flux elicited by C9ORF72 expression.

Minor concerns that should be addressed

- 1) most of the images shown are not very convincing. In particular, images in Figure 1a do not appear to illustrate a doubling of EGFP puncta between the vehicle and torin condition in control conditions. Moreover, some cells (eg C9si+torin cell in Fig1a or ctrl/bafA1 cell in Fig1b) but not others show a distinctive nuclear staining. The authors should precise how they standardized

exposure conditions and all quantifications should be performed in blinded manner.
2) in figures 1c and d, the experiments should include conditions without bafilomycin similar to figure 1a.

Referee #2:

GGGGCC repeat expansions in the first intron of C9ORF72 are a common cause of both ALS and FTD. A potential pathogenic mechanism is partial loss of C9ORF72. Thus, understanding the normal function of C9ORF72 is critically important. To this end, Webster and colleagues demonstrated that C9ORF72 helps initiate autophagy and that loss of C9ORF72 results in the formation of p62 puncta in proliferating tumor cells. These points were already reported very recently by Sellier et al. (EMBO J. April 21, 2016). Understandably, Webster and colleagues may feel disappointed because they are not the first to report this finding. On the other hand, it is nice to see different groups reach the same conclusion independently. Another weakness of the current study is that Webster and colleagues used proliferating tumor cell lines such as HeLa cells. It would be better to perform at least some experiments in primary neurons, as Sellier et al. did. This is important, especially considering the recently reported cell-type-specific phenotypes in C9orf72 KO mice (Atanasio et al., 2016; O'Rourke et al., 2016). To enhance the significance and novelty of their study, could the authors obtain published C9orf72 KO mice and examine the autophagy pathway in neurons and microglia? Although both Sellier et al. and Webster et al. reported a role for C9ORF72 in autophagy initiation, the detailed mechanisms revealed by these two groups are different. Webster et al. found that C9ORF72 binds to RAB1, as reported by Farg et al. (HMG 2014). In contrast, Sellier et al. identified a novel C9ORF72 complex that interacts with Rab8a and Rab39b but not RAB1. Webster et al. concluded that C9ORF72 is an effector of RAB1, but Sellier et al. found that the C9ORF72 complex functions as a GEF for RAB39b. Webster et al. found C9ORF72 recruits ULK1 to phagophores, while Sellier et al. showed that a subunit of the C9ORF72 complex is phosphorylated by ULK1. These differences should be resolved. In the current study, the authors also reported compromised autophagy in patient neurons, consistent with a previous report (Almeida et al., 2013). On a technical note, at least some if not all the siRNA knockdown studies should include rescue experiments to demonstrate the cellular phenotypes are not due to off-target effects.

1st Revision - authors' response

23 May 2016

Referee #1:

1) Most of the studies were done in cell lines that are not even neuronal (HEK293 and HeLa cells). Autophagy is notoriously important in neurons, and likely to be regulated by different pathways/effectors than in peripheral cells. Thus, the key data should be reproduced in (ideally primary) neuronal cells. It is worth mentioning that some data are reproduced in induced neurons, but this is not the case of Rab1a or FIP200 involvement.

Response:

We have now included substantial new data that reproduce the key findings in primary cortical neurons as well as SH-SY5Y neuroblastoma cells:

- In Fig 1E and D we show that miRNA-mediated knockdown of C9orf72 in primary rat cortical neurons impairs initiation of autophagy.
- In Fig 4C we show that miRNA-mediated knockdown of C9orf72 in primary rat cortical neurons impairs Torin1-induced translocation of FIP200, and that this effect can be rescued by re-expressing human C9orf72.
- In Fig 5B we show in SH-SY5Y neuroblastoma cells that siRNA knockdown of Rab1a impairs translocation of FIP200 induced by overexpression of C9orf72.
- In Fig 5C we show in SH-SY5Y neuroblastoma cells that siRNA knockdown of Rab1a impairs induction of autophagy induced by overexpression of C9orf72.
- In Fig 8B we show that knockdown of C9orf72 in rat primary cortical neurons causes p62 accumulation and that this effect can be rescued by re-expressing human C9orf72.

Thus, we believe that these new data convincingly show that C9orf72 regulates initiation of autophagy in primary neurons and neuronal cell lines.

2) the interaction between C9ORF72 and the ULK1 complex remains poorly documented. The authors perform 2 sets of experiments, each with their weaknesses. First co-Ips are performed only on transfected, overexpressing cells. While C9ORF72 are notoriously poor, the authors could at least show that endogenous FIP200, ULK1 and Rab1a interact with transfected C9ORF72 isoforms. In the same line, it is not clear while some non specific bands are present only in GST-C9ORF72 wells in Figure 3B, and not in the GST control. It seems also that C9ORF72L is present in the recombinant GST-C9ORF72S preparation ? To document a true interaction, the authors should be more quantitative, and perform competition assays, vary salinity conditions, or, even better measure interaction constants with BiaCore or similar setup. Alternatively, and even if very indirect, a PLA assay between C9ORF72 and Ulk1 members would bring additional evidence. As such, I am not convinced that C9ORF72 and members of the ULK1 complex are true interactors in vivo.

Response:

As suggested by the Referee we have now included endogenous immunoprecipitations, proximity ligation assays, and equilibrium binding assays that confirm the interaction of C9orf72 with ULK1, FIP200, ATG13 and Rab1a:

- In Figs 3D and E, we show that endogenous FIP200, ULK1 and ATG13 co-immunoprecipitate with transfected C9orf72.
- In Figs 3F, G and H, we show that FIP200, ULK1 and ATG13 interact with C9orf72 in PLA assays.
- In Fig 6A we show the Y2H assay in which we identified Rab1a as a C9orf72 binding partner.
- In Fig 6C we show that endogenous Rab1a co-immunoprecipitates with transfected C9orf72.
- In Fig 6E we show new equilibrium binding assays in which we use GST-Rab1 to pull down *in vitro* translated, radiolabeled C9orf72. These assays show that the interaction of C9orf72 with Rab1a fits a hyperbola consistent with specific biological interaction.

We identified the non-specific bands in the GST-C9orf72 wells the Referee refers to as DnaK Chaperonin E. coli, and CH60 E. coli - 60 kDa Chaperonin – these data were originally shown in Appendix Fig S2 (now S3). We always find these contaminating proteins, even when increasing ATP concentrations to dislodge chaperones.

DnaK Chaperonin E. coli runs just below GST-C9orf72L and this gives the impression on the coomassie gels that there is GST-C9orf72L present in the GST-C9orf72S samples – in fact it is DnaK Chaperonin E. coli. This is indicated in the figure legends of Figs 3 and 6.

To exclude possible effects of the chaperones on the interaction of C9orf72 with its binding partners we have now performed the reverse binding assay where we use GST-Rab1 (without contaminating chaperones) to pull down *in vitro* translated, radiolabeled C9orf72 (Fig 6E).

Together with the original data these findings show that the interaction between C9orf72, the ULK1 complex and Rab1a are true interactions.

3) the specificity of the knockdown of C9ORF72 effects could be shown to be rescued by a siRNA resistant form of C9ORF72. This is especially important in induced neurons as these neurons also have potential RNA toxicity and DPR toxicity, that could in principle also interfere with autophagy.

Response:

At the suggestion of the Referee we have addressed the specificity of the C9orf72 knock down in two ways in accordance with published guidelines (2003). First we have now included new experiments in which we functionally rescue knockdown of C9orf72:

- In Fig 4C we show that re-expressing human C9orf72 rescues Torin1-induced translocation of FIP200 in rat cortical neurons in which rat C9orf72 was knocked down with miRNA.
- In Fig 8B we show that re-expressing human C9orf72 rescues p62 accumulation in rat cortical neurons in which rat C9orf72 was knocked down with miRNA.

Secondly we have now included a multiplicity control showing that the two C9orf72 siRNAs that target different C9orf72 sequences and were used as pool in our study (see Materials and Methods) elicit the same effects on autophagy when used separately (Appendix Fig S1).

We think that given the evidence in primary neurons presented in the revised manuscript it is most likely that the autophagy deficits in iNeurons are due to loss of C9orf72. However, we agree with

the Referee that there remains a possibility that RNA and DPR toxicity also interfere with autophagy. Regrettably, we were not able to perform rescue experiments in iNeurons because of the short 4-week limit on this revision. While this is unfortunate, it does not detract from the main message of our manuscript. We want to emphasize that the goal of this study was foremost to characterize the function of C9orf72 and we feel that extensive characterization of RNA toxicity and DPRs and their possible effects on autophagy and other pathways in the iNeurons should be addressed in future studies.

4) the authors show that overexpression of constitutively active Rab1a reverts the effects on FIP200 punctae. This suggests that Rab1a activation is sufficient to revert this defect due to C9ORF72 knock down. However, and most importantly, the authors do not document the effects on LC3 and autophagosome formation in the same experimental conditions (although the magnitude of the expected effects is quite larger - compare Figure 1b with Figure 4b). It would also be important to perform the converse experiment, that is to show that Rab1a dominant negative expression inhibits increased autophagic flux elicited by C9ORF72 expression.

Response:

We believe the Referee has misinterpreted our constitutively active Rab1a data. In the original Fig 5D (now Fig 7B) we show that dominant active (DA) Rab1a drives FIP200 translocation and that this is inhibited by knockdown of C9orf72. This is consistent with a model in which C9orf72 recruits Rab1a to the ULK1 complex to drive translocation. This model is also supported by the PLA data in Fig 7A (original Fig 5C) in which we show that the interaction between Rab1a and ULK1 is lost in C9orf72 knockdown cells. Accordingly, constitutively active Rab1a does not revert the effects on FIP200 puncta as the Referee asserts, but C9orf72 is required for Rab1a to drive translocation of the ULK1 complex. We apologize if this was not clear.

We agree with the Referee that the effects on LC3 are worth documenting and we have now included these data in Fig 7C. The data show that DA Rab1a increases autophagosome formation and that knockdown of C9orf72 inhibits this effect. Again this is consistent with C9orf72 recruiting Rab1a to the ULK1 complex.

The Referee asks to show that Rab1a dominant negative expression inhibits increased autophagic flux elicited by C9orf72 expression. We have now included these data in Extended View Fig EV3. Fig EV3A shows that dominant negative Rab1a inhibits increased FIP200 translocation elicited by C9orf72 expression; this confirms the data in Fig 5A (original Fig 4D) showing the same but using Rab1a siRNA instead of DN Rab1a. Fig EV3B shows that dominant negative Rab1a inhibits increased autophagic flux elicited by C9ORF72 expression, and Fig 5C shows the same but using Rab1a siRNA.

Minor concerns that should be addressed

1) most of the images shown are not very convincing. In particular, images in Figure 1a do not appear to illustrate a doubling of EGFP puncta between the vehicle and torin condition in control conditions. Moreover, some cells (eg C9si+torin cell in Fig1a or ctrl/bafA1 cell in Fig1b) but not others show a distinctive nuclear staining. The authors should precise how they standardized exposure conditions and all quantifications should be performed in blinded manner.

The images shown in Fig 1A have Ctrl:78 and Torin1:108 EGFP puncta, respectively. While not exactly a doubling, there is a clear increase in LC3 puncta in the Torin1 condition. As there is a standard deviation on the mean number of puncta per cell this is not unexpected. We do agree we could have chosen an image that more accurately represented the mean and we have now replaced the Ctrl image with a new one containing 36 LC3 puncta.

When EGFP-LC3 and mCherry-EGFP-LC3 is transfected there always is nuclear staining – this is consistent with the pool of nuclear LC3 that has been reported by several labs (Drake *et al.*, 2010; Dou *et al.*, 2015; Huang *et al.*, 2015). We don't think this detracts from the results as we focused on cytoplasmic puncta. The exposure times, excitation intensity and camera settings were kept constant for all samples in an experiment and the quantifications were performed blinded. This was indicated in the original Material and Methods section.

2) in figures 1c and d, the experiments should include conditions without bafilomycin similar to figure 1a.

The full blots with all conditions are now included in Extended View Fig EV1. We opted to only show the bafilomycin-treated conditions in Fig 1 because we were specifically testing induction of autophagy.

Referee #2:

GGGGCC repeat expansions in the first intron of C9ORF72 are a common cause of both ALS and FTD. A potential pathogenic mechanism is partial loss of C9ORF72. Thus, understanding the normal function of C9ORF72 is critically important. To this end, Webster and colleagues demonstrated that C9ORF72 helps initiate autophagy and that loss of C9ORF72 results in the formation of p62 puncta in proliferating tumor cells. These points were already reported very recently by Sellier et al. (EMBO J. April 21, 2016). Understandably, Webster and colleagues may feel disappointed because they are not the first to report this finding. On the other hand, it is nice to see different groups reach the same conclusion independently. Another weakness of the current study is that Webster and colleagues used proliferating tumor cell lines such as HeLa cells. It would be better to perform at least some experiments in primary neurons, as Sellier et al. did. This is important, especially considering the reported cell-type-specific phenotypes in C9orf72 KO mice (Atanasio et al., 2016; O'Rourke et al., 2016). To enhance the significance and novelty of their study, could the authors obtain published C9orf72 KO mice and examine the autophagy pathway in neurons and microglia?

Response:

We agree with the Referee and have now included substantial new data that reproduce the key findings in primary cortical neurons as well as SHSY-5Y neuroblastoma cells (see response to Referee #1). These new data convincingly show that C9orf72 regulates initiation of autophagy in primary neurons and neuronal cell lines.

We appreciate the Referee's comments and agree that examining the autophagy pathway in neurons and microglia from published C9orf72 KO mice would be an interesting line of research. Presently we don't have C9orf72 KO mice in our lab, so we could not prepare primary neurons or microglia for this revision. Hence we have opted to knockdown C9orf72 in primary neurons.

We have obtained some embryonic fibroblasts from an as yet unpublished C9orf72 KO mouse from the Pasterkamp lab in Utrecht and we find that there is a defect in autophagy in these cells. We also have performed autophagy assays in an unpublished zebrafish knockout model of C9orf72 and again found an autophagy defect. Since we only tested the fibroblasts once and are still in the process of characterizing the zebrafish, we feel that these data are too preliminary to be included in the present study. (Figures for Referees not shown.)

Although both Sellier et al. and Webster et al. reported a role for C9ORF72 in autophagy initiation, the detailed mechanisms revealed by these two groups are different. Webster et al. found that C9ORF72 binds to RAB1, as reported by Farg et al. (HMG 2014). In contrast, Sellier et al. identified a novel C9ORF72 complex that interacts with Rab8a and Rab39b but not RAB1. Webster et al. concluded that C9ORF72 is an effector of RAB1, but Sellier et al. found that the C9ORF72 complex functions as a GEF for RAB39b. Webster et al. found C9ORF72 recruits ULK1 to phagophores, while Sellier et al. showed that a subunit of the C9ORF72 complex is phosphorylated by ULK1. These differences should be resolved.

Response:

We agree it will be important to further investigate the molecular mechanisms to try and build a model that reconciles both our findings and those of Sellier et al., but feel this is outside the scope of the current manuscript.

In the current study, the authors also reported compromised autophagy in patient neurons, consistent with a previous report (Almeida et al., 2013). On a technical note, at least some if not all the siRNA knockdown studies should include rescue experiments to demonstrate the cellular phenotypes are not due to off-target effects.

Response:

As mentioned in our response to Referee#1, we have addressed possible off-target effects by functional rescue as suggested by the Referees and by a multiplicity control experiment in accordance with published guidelines (2003). In Figs 4C and 8B we show that re-expressing human C9orf72 rescues C9orf72 knockdown in rat cortical neurons, and in Appendix Fig S1 we show that the two C9orf72 siRNAs that target different C9orf72 sequences and were used as pool in our study elicit the same effects on autophagy when used separately. For the Rab1a knockdown experiments we have also used dominant negative Rab1A to inhibit Rab1a and found the same results (Fig EV3).

References:

- Dou Z, Xu C, Donahue G, Shimi T, Pan JA, Zhu J, Ivanov A, Capell BC, Drake AM, Shah PP, Catanzaro JM, Ricketts MD, Lamark T, Adam SA, Marmorstein R, Zong WX, Johansen T, Goldman RD, Adams PD, Berger SL (2015) Autophagy mediates degradation of nuclear lamina. *Nature*, **527**: 105–109
- Drake KR, Kang M, Kenworthy AK (2010) Nucleocytoplasmic distribution and dynamics of the autophagosome marker EGFP-LC3. *PLoS One*, **5**: e9806
- Huang R, Xu Y, Wan W, Shou X, Qian J, You Z, Liu B, Chang C, Zhou T, Lippincott-Schwartz J, Liu W (2015) Deacetylation of nuclear LC3 drives autophagy initiation under starvation. *Mol Cell*, **57**: 456–466
- (2003) Whither RNAi. *Nat Cell Biol*, **5**: 489–490

Knockout of C9orf72 in mice and zebrafish impairs autophagy.

- A. Mouse embryonic fibroblasts (MEFs) from wild type (Ctrl) and C9orf72 knockout (KO) mice were treated with vehicle, Torin1, bafilomycin A1 (BafA1) or Torin1+BafA1 and processed for immunoblot analysis of LC3 conversion. Torin1 induced autophagy in Ctrl MEFs whereas in C9orf72 KO MEFs there is no induction (compare LC3-II, BafA1 v Torin1+BafA1).
- B. Wild type and C9orf72 KO zebrafish embryos were treated with NH₄Cl (equivalent to BafA1) or NH₄Cl+Torin1 and processed for immunoblot analysis of LC3 conversion. Torin1 induced autophagy in wild type but not C9orf72 KO zebrafish.

2nd Editorial Decision

01 June 2016

Thank you for submitting your manuscript to the EMBO journal. Referee #1 has re-viewed your revision and as you can see below the referee appreciates the introduced changes. There is one remaining point that should be addressed namely to cite and discuss the two recent and related studies on this topic (Sellier et al, 2016; Sullivan et al., 2016). You can use the link below to upload the revised version.

REFEREE REPORTS

Referee #1:

The manuscript by Webster and colleagues has been profoundly revised and addresses now most of my initial comments. This is a very important effort in such a short time frame.

In particular, the confirmation of the effects in neuronal cells, the rescue of some of the siRNA effects by overexpression of C9ORF72, are important controls that are extremely important and are now provided.

A minor point that still needs to be addressed is the comparison between this study and two other papers that recently appeared and are not cited: Sellier et al, 2016; Sullivan et al., 2016. I think the difference in the Rab identified should be discussed, as well as the identity of the other members of the C9ORF72 complex (SMCR8 and WDR41).

Referee #1:

The manuscript by Webster and colleagues has been profoundly revised and addresses now most of my initial comments. This is a very important effort in such a short time frame. In particular, the confirmation of the effects in neuronal cells, the rescue of some of the siRNA effects by overexpression of C9ORF72, are important controls that are extremely important and are now provided.

We appreciate that the Referee #1 recognizes our efforts and thank him/her again for the helpful comments.

A minor point that still needs to be addressed is the comparison between this study and two other papers that recently appeared and are not cited: Sellier et al, 2016; Sullivan et al., 2016. I think the difference in the Rab identified should be discussed, as well as the identity of the other members of the C9ORF72 complex (SMCR8 and WDR41).

We apologize for this oversight and have now discussed and cited both papers in the Discussion section on page 11.

Corresponding Author Name: Kurt De Vos

Journal Submitted to: EMBOJ

Manuscript Number: EMBOJ-2016-94401